# Synchronization of oscillatory growth prepares fungal hyphae for fusion

**Valentin Wernet[1], Marius Kriegler[1], Vojtech Kumpost[2,3], Ralf Mikut[2], Lennart Hilbert[3,4], Reinhard Fischer[1]***

[1]Karlsruhe Institute of Technology - South Campus Institute for Applied Biosciences Dept. of Microbiology, Karlsruhe, Germany; [2]Karlsruhe Institute of Technology – North Campus Institute for Automation and Applied Informatics, Eggenstein-Leopoldshafen, Germany; [3]Karlsruhe Institute of Technology – North Campus Institute of Biological and Chemical Systems – Biological Information Processing, Eggenstein-Leopoldshafen, Germany; [4]Karlsruhe Institute of Technology – South Campus Zoological Institute Dept. of Systems Biology / Bioinformatics, Eggenstein-Leopoldshafen, Germany

**\*For correspondence:**
reinhard.fischer@kit.edu

**Competing interest:** The authors declare that no competing interests exist.

**Abstract** Communication is crucial for organismic interactions, from bacteria, to fungi, to humans. Humans may use the visual sense to monitor the environment before starting acoustic interactions. In comparison, fungi, lacking a visual system, rely on a cell-to-cell dialogue based on secreted signaling molecules to coordinate cell fusion and establish hyphal networks. Within this dialogue, hyphae alternate between sending and receiving signals. This pattern can be visualized via the putative signaling protein Soft (SofT), and the mitogen-activated protein kinase MAK-2 (MakB) which are recruited in an alternating oscillatory manner to the respective cytoplasmic membrane or nuclei of interacting hyphae. Here, we show that signal oscillations already occur in single hyphae of *Arthrobotrys flagrans* in the absence of potential fusion partners (cell monologue). They were in the same phase as growth oscillations. In contrast to the anti-phasic oscillations observed during the cell dialogue, SofT and MakB displayed synchronized oscillations in phase during the monologue. Once two fusion partners came into each other's vicinity, their oscillation frequencies slowed down (entrainment phase) and transit into anti-phasic synchronization of the two cells' oscillations with frequencies of 104±28 s and 117±19 s, respectively. Single-cell oscillations, transient entrainment, and anti-phasic oscillations were reproduced by a mathematical model where nearby hyphae can absorb and secrete a limited molecular signaling component into a shared extracellular space. We show that intracellular $Ca^{2+}$ concentrations oscillate in two approaching hyphae, and depletion of $Ca^{2+}$ from the medium affected vesicle-driven extension of the hyphal tip, abolished the cell monologue and the anti-phasic synchronization of two hyphae. Our results suggest that single hyphae engage in a 'monologue' that may be used for exploration of the environment and can dynamically shift their extracellular signaling systems into a 'dialogue' to initiate hyphal fusion.

## Editor's evaluation

This important study combines convincing live cell imaging and mathematical modeling data to show how an emerging model fungus engages in an oscillatory chemical dialogue to prepare for cell-cell fusion. In the absence of a fusion partner, fungal hyphae undergo signal oscillations that are in phase with their growth oscillations; following detection of a fusion partner, the oscillation frequencies slow down, and a transition to an anti-phasic synchronization of the oscillations between the two partners takes place. Based on a mathematical model, the authors suggest a mechanism involving the oscillatory secretion/uptake of a signaling compound from a shared extracellular space.

## Introduction

Oscillations are common phenomena in biology with periods from seconds to hours, to days, to years (*Damineli et al., 2022*; *in 't Zandt et al., 2021*; *Dunlap and Loros, 2017*). In fungi, oscillations are a well-described feature of hyphal tip extension, where calcium ions control vesicle accumulation, actin depolymerization, and vesicle fusion with the tip membrane (*Takeshita et al., 2017*). In addition, signal oscillations have been described at the hyphal tips during the fusion of hyphal cells of *Neurospora crassa, Fusarium oxysporum, Botrytis cinerea*, and *Arthrobotrys flagrans* (*Fischer et al., 2018*; *Fleißner and Herzog, 2016*; *Fleissner et al., 2009*; *Youssar et al., 2019*; *Palos-Fernández et al., 2022*; *Roca et al., 2012*). These signal oscillations were named *cell-to-cell dialogue* and are probably based on a conserved diffusible signaling molecule, which remains to be discovered (*Goryachev et al., 2012*; *Haj Hammadeh et al., 2022*; *Daskalov et al., 2017*). The nematode-trapping fungus *A. flagrans* can switch from a saprotrophic to a predatory lifestyle, and sophisticated signaling regimes between the fungus and its prey, *Caenorhabditis elegans,* control trap formation, attraction of the prey, and attack of the nematode (*Hsueh et al., 2017*; *Yu et al., 2021*; *Wernet et al., 2021*; *Fischer and Requena, 2022a*). Once captured *A. flagrans* penetrates the cuticle of the nematode and colonizes the worm body. There is first evidence that small-secreted proteins play important roles in the fungal attack (*Youssar et al., 2019*; *Wernet et al., 2021*; *Fischer and Requena, 2022b*). Nematode-trapping fungi have also the potential to be applied as biocontrol agents against animal and plant-pathogenic nematodes (*Wernet and Fischer, 2023*; *Rodrigues et al., 2022*).

Two of the hallmark proteins during the cell-to-celll dialogue of fungal hyphae are the *soft* protein, SofT, and the mitogen-activated protein (MAP) kinase MakB which show anti-phasic, oscillatory recruitment to the plasma membrane and the nuclei of interacting cells, respectively (*Fleissner et al., 2009*; *Haj Hammadeh et al., 2022*). Although its molecular function remains elusive, SofT is thought to be involved in generating a signal during the cell-to-cell dialogue. It is essential for cell fusion in filamentous fungi and acts as a scaffold protein of the cell wall integrity pathway (*Fischer and Glass, 2019*; *Teichert et al., 2014*; *Fleissner et al., 2005*; *Serrano et al., 2022*). Besides the nature of the signaling compound, another important open question is the onset and coordination of the communication process. If people want to communicate, they may use their vision to approach each other, before the start of an acoustic conversation. However, if now the scene changes to a dark room where visual information is not available, the initiation and coordination of the conversation appears to be difficult. As the cell-to-cell dialogue in fungal hyphae appears to be based on a single, chemical communication channel, it is proposed that the two involved cells undergo oscillatory secretion of a signaling compound with a refractory period that prevents self-stimulation and can contribute to overcoming critical threshold concentrations (*Damineli et al., 2022*). Here, we investigated the onset and coordination of that cell-to-cell dialogue and found that single hyphae show the same signaling events at the hyphal tip as during the cell-to-cell dialogue. Once hyphae meet, the 'monologue' of each hypha transits into a dialogue.

## Results

An open question is if and how cell-to-cell communication is activated once a hyphal fusion partner appears in the vicinity. To address this question, we monitored SofT-GFP in *A. flagrans* growing on low-nutrient agar (LNA) and observed oscillatory recruitment of SofT to single hyphal tips with a mean period of 137±17 (SD) seconds (*n*=164 in 18 hyphae) without other hyphae in their vicinity (*Figure 1a, d, and e*, *Video 1*). We measured SofT-GFP signals at hyphal tips and found that 85 ± 4% (mean ± SD) showed fluorescence without a fusion partner (*n*=50) hyphal tips, experiment repeated three times, (*Figure 1—figure supplement 1a*). Hyphae without SofT-GFP recruitment at their tips were unable to fuse, suggesting that they could not recognize each other (*Figure 1—figure supplement 1b*, *Video 2*). However, these cells were still able to undergo cell fusion if other hyphae induced hyphal fusion (*Figure 1—figure supplement 1b*). Recruitment of SofT-GFP to the hyphal tip and cell fusion were also observed on potato dextrose agar (PDA), suggesting that nutrient availability would not be the sole cause of inducing the observed dynamics (*Figure 1—figure supplement 1c*). The observed SofT-GFP dynamics resembled oscillating tip growth described in *Aspergillus nidulans* and raised the question whether and how these two processes might be connected in *A. flagrans* (*Takeshita et al., 2017*). Therefore, we monitored the mCherry-tagged orthologue of *A. nidulans* chitin synthase B

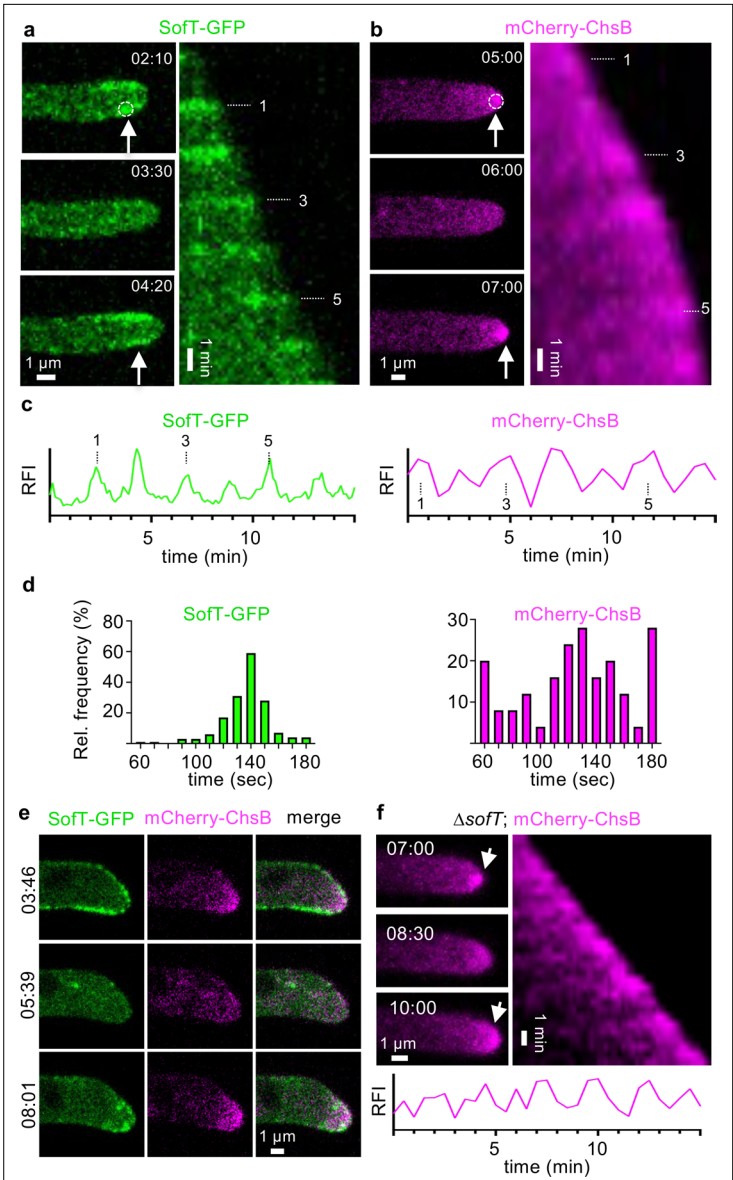

**Figure 1.** Oscillatory recruitment of signaling proteins during hyphal growth of *A. flagrans*. (**a, b**) Time course of SofT-GFP and mCherry-ChsB localization at hyphal tips. Arrows indicate the accumulation of the proteins at the hyphal tips. Dotted circles indicate an area of 6×6 pixels which was used to measure the fluorescent intensity over time depicted in (**c**) of GFP-SofT at the lower edge of the plasma membrane or of mCherry-ChsB at the apex. (**b**) is a maximum-intensity projection of the time-lapse sequence. A kymograph was created for each time course by drawing a line (pixel width 5) along the growth axis of the respective hypha. The numbers represent the count of oscillating accumulations of each fusion protein during the growth of each hypha in the corresponding time course. (**c**) Relative fluorescent intensity (RFI, y-axis, arbitrary units) at the hyphal tips of a–b was measured over the time course (x-axis in minutes). (**d**) The interval between two peaks at hyphal tips was counted and depicted as relative frequency (y-axis) over the time (in seconds). GFP-SofT *n*=164 in 8 hyphae. mCherry-ChsB *n*=50 in 7 hyphae. (**e**) Localization of GFP-SofT (depicted in green) and mCherry-ChsB (depicted in magenta) during hyphal tip growth. Numbers indicate the time in minutes. (**f**) Localization of ChsB-mCherry at the hyphal tip of the *A. flagrans* Δ*sofT*-mutant strain. The relative fluorescent intensity at the hyphal tip (y-axis, arbitrary units) was measured over the time course.

The online version of this article includes the following figure supplement(s) for figure 1:

**Figure supplement 1.** Localization of SofT-GFP at hyphal tips of *A. flagrans*.

**Figure supplement 2.** The N-terminal disordered region and WW domain of SofT are sufficient to perform vegetative hyphal fusion in *A. flagrans*.

Localization of
GFP-SofT during hyphal tip
growth.

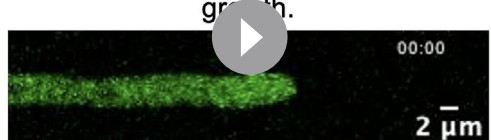

**Video 1.** Localization of GFP-SofT during hyphal tip growth.
https://elifesciences.org/articles/83310/figures#video1

Time course of
GFP-SofT during a
hyphal fusion event.

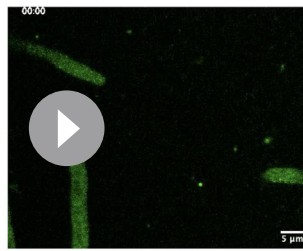

**Video 2.** Time course of GFP-SofT during a hyphal fusion event.
https://elifesciences.org/articles/83310/figures#video2

(DFL_009443) in *A. flagrans*, which acts as a cargo marker for intracellular transport and is crucial for polarized growth (*Zhou et al., 2018*; *Hernández-González et al., 2018*). The fluorescent fusion protein oscillated at growing hyphal tips with a mean period of 125±37 (SD) seconds (*n*=50 in 7 hyphae) (*Figure 1b, d, and e*, *Video 3*). To compare the oscillations of SofT and ChsB, we created a strain with both proteins labeled. SofT-GFP localized at the same time at growing tips as mCherry-ChsB (*Figure 1e*, *Video 4*). However, mCherry-ChsB showed a wider frequency distribution and shorter periods. Thus, the localization of mCherry-ChsB appears to be independent of SofT (*Figure 1f*). Indeed, localization and oscillation of mCherry-ChsB at the tip were not affected by deletion of *sofT* (*Figure 1f*). Deletion of *chsB* in *A. nidulans* is detrimental for hyphal growth and was therefore not analyzed in *A. flagrans* (*Borgia et al., 1996*; *Fukuda et al., 2009*). To further analyze its molecular role, we generated three truncated versions of the *A. flagrans* SofT protein and performed rescue experiments of the ΔsofT mutant (*Figure 1—figure supplement 2*, *Youssar et al., 2019*). SofT consists of 1213 amino acids with a predicted N-terminal disordered region (AA1–540), a WW domain (AA505–533), and a putative C-terminal phosphatase domain (AA676–1213) (*Figure 1—figure supplement 2a*). A fragment containing the N-terminal region and the WW domain was sufficient to restore aerial mycelia formation and hyphal fusion, while the N-terminal disordered region or the C-terminus was not, emphasizing the significance of SofT-interacting proteins in cell communications (*Figure 1—figure supplement 2b, c*). Our results show that most *A. flagrans* hyphae are constantly sending signals in a form of constant 'self-talk', or monologue, possibly to explore the environment for a fusion partner.

To understand how the signal oscillations in a single hypha (monologue) might transit into the anti-phasic, cell-to-cell dialogue once a partner cell appears in the vicinity, we constructed a mathematical model. Specifically, we extended an existing model of cell-to-cell communication based on anti-phasic oscillations so as to explicitly account for the uptake of signaling components from the surrounding media (*Goryachev et al., 2012*). This model requires that not one, but two species in the cell undergo a conversion, with the second species' conversion requiring the first species' transition to have completed to a large extent. The reason behind this is that, without this sequential conversion, there is no delay between receiving and secreting signal, and that the secreting cell would not become non-receptive to its own signal (*Goryachev et al., 2012*). The concentration of signaling components in the immediate proximity of the wall of cell 1 and cell 2 is represented by two model variables, $Z_1$ and $Z_2$, respectively (*Figure 2a*, *Figure 2—figure supplement 1*). As the signaling component is taken up into a cell, activating components

Localization of
mCherry-ChsB during
hyphal tip growth.

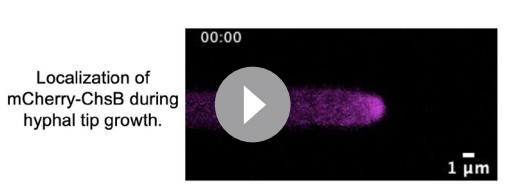

**Video 3.** Localization of mCherry-ChsB during hyphal tip growth.
https://elifesciences.org/articles/83310/figures#video3

Co-localization of
GFP-SofT (depicted in green)
and mCherry-ChsB (depicted in
magenta) during hyphal tip
growth.

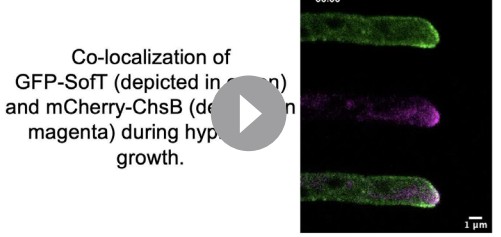

**Video 4.** Co-localization of GFP-SofT (depicted in green) and mCherry-ChsB (depicted in magenta) during hyphal tip growth.
https://elifesciences.org/articles/83310/figures#video4

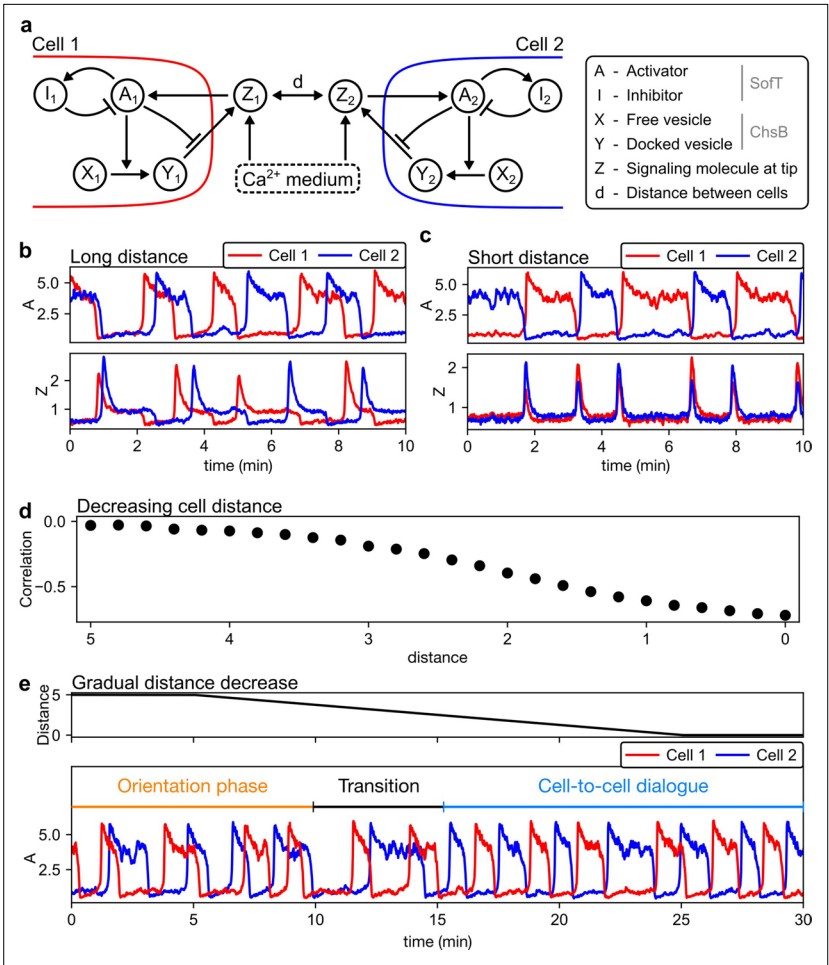

**Figure 2.** A mathematical model demonstrates the emergence of the temporal synchronization of two hyphae approaching each other. (**a**) In the model, each hyphal cell contains an excitatory system with an activator ($A_{1,2}$) and inhibitor ($I_{1,2}$). This excitatory system is triggered by extracellular signaling molecules ($Z_{1,2}$) and, in turn, regulates the docking and subsequent release of vesicles ($X_{1,2}$ and $Y_{1,2}$), which contain the signaling protein, into the extracellular space. (**b**) At long distances (here, $d = 10$), the cells operate as independent oscillator units. (**c**) At short distances (here, $d = 0$), cells synchronize and oscillate in anti-phase. (**d**) Decreasing the distance between the two cells increases the magnitude of the anti-correlation (negative Pearson correlation coefficient) between the activator concentrations of the individual cells. (**e**) Upon continuous reduction of the distance between the cells, anti-synchrony emerges (noticeable at around 15 min).

The online version of this article includes the following figure supplement(s) for figure 2:

**Figure supplement 1.** Example simulation containing a single cell to illustrate the oscillatory dynamics of all model variables.

(modeled as variables $A_1$ and $A_2$) become more concentrated in this cell. These activating components, in turn, stimulate the docking of cytoplasmic signaling components ($X_1$ and $X_2$) to the inside of the cell membrane ($Y_1$ and $Y_2$). The release of membrane-docked vesicles to the extracellular space is initially blocked by high levels of activator ($A_1$ and $A_2$). Secretion ultimately occurs when some of the activating components are converted into an auto-inhibitory component ($I_1$ and $I_2$), reducing levels of activating components, thereby allowing secretion of signaling components into the extracellular compartments ($Z_1$ and $Z_2$). The main novelty of our model is the addition of the variables $Z_1$ and $Z_2$ to explicitly represent the information exchange between hyphae via the shared extracellular medium. Among the abstract model variables, SofT as a signaling component located inside of a hyphal cell connects most closely to variables $X_{1,2}$ and $Y_{1,2}$ (**Figure 2a**). ChsB is also located inside the hyphal cells, but localizes differently from SofT and likely does not act as a signaling component, so that ChsB can be related to the variables $A_{1,2}$, and $I_{1,2}$.

In simulations containing only a single cell, similar to the experimental results monitoring the components in a single hypha, oscillations in these different components could be observed (*Figure 1—figure supplement 1*). When placing two cells in the simulation at different distances ($d$) from each other, oscillations of these two cells appear uncoordinated at long distances (monologue), and anti-phasic at short distances (dialogue) (*Figure 2b–d*). The role of the signaling component in the dialogue at a short distance can be seen in our simulations: high extracellular concentrations ($Z_1$ and $Z_2$) are attained only briefly, when one cell has secreted the component, and the other cell has not yet taken it up again (*Figure 2c*). This back-and-forth exchange of signaling component via secretion into and uptake from the extracellular space represents a physically credible mechanism to establish the cell-to-cell dialogue at a short distance. When we implement a simulation that mimics the growth of two hyphae toward each other in the form of a distance that decreases over time, a transitory phase where the uncoordinated oscillations (two monologues) become mutually entrained into anti-phasic oscillations (dialogue) can be seen (*Figure 2e*). As seen in these simulation results, the same regulatory mechanism, based on a signaling component that is exchanged via the extracellular space, can explain the transition from single-cell oscillatory growth to anti-phasic synchronization between two approaching cells. Crucially, these simulations also imply a transitory phase, during which both cells' dynamics slow down, just before entering into the dialogue and speeding up again (*Figure 2e*). This 'critical slowing down' is typical in phases where a system transitions from one type of dynamic behavior to another type of dynamic behavior, due to being 'caught in between' two types of behavior during the transition (*Damineli et al., 2022*; *Quail et al., 2015*).

To experimentally assess the transition from two 'monologues' to a coordinated 'dialogue', we monitored SofT in two approaching hyphae. Initially, the oscillations in the two hyphae appeared uncoordinated (orientation phase) but then transitioned to anti-phasic oscillations characteristic of the cell-to-cell dialogue (*Figure 3a and b*, *Video 5*). The frequency of SofT oscillations during the cell-to-cell dialogue (mean period of 104±28 [SD] seconds, $n$=173 in 18 hyphae) was comparable to the frequency in single hyphae and was not influenced by the presence of nematodes (*Figure 3c*). As expected from our simulations of a dynamic transition into coordinated oscillations, oscillations in both hyphae slowed down during the transitory phase that precedes the coordinated cell-to-cell dialogue (*Figure 3b*). To further validate the findings obtained with SofT, we investigated the behavior of the MAP kinase MakB, which is central for the cell dialogue and exhibits anti-phasic oscillations with SofT (*Haj Hammadeh et al., 2022*). We monitored the localization of MakB tagged with mCherry and observed oscillatory recruitment of MakB to the nuclei of hyphae with a mean period of 130±33 ($n$=45 in 8 hyphae) in the absence of neighboring hyphae and a mean period of 117±19 ($n$=57 in 8 hyphae) during the cell-to-cell dialogue (*Figure 3c*, *Figure 3—figure supplement 1*, *Video 6*). Interestingly, co-localization of SofT-GFP and MakB-mCherry in the same hyphae revealed that both proteins were oscillating in the same phase without other hyphae in their vicinity (*Figure 3d and e*, *Video 7*), which is opposite to the so far observed anti-phasic oscillations observed during the cell dialogue (*Fleissner et al., 2009*; *Haj Hammadeh et al., 2022*). Additionally, we observed that decoupling of SofT-GFP and MakB-mCherry oscillations into the anti-phasic cell dialogue occurred during the transitory phase where the growth slowed down (*Figure 3f*, *Video 8*). These results indicate that the transition from a 'monologue' to a 'dialogue' include the decoupling of SofT and MakB oscillations.

The underlying hypothesis that the same oscillation mechanism acts during the monologue- and dialogue-type dynamics is further substantiated by the observation that, similar to single hyphal growth, not only SofT and MakB but also ChsB and actin filaments visualized by Lifeact-GFP showed oscillating recruitment to the tips of interacting cells with mean periods of 103±26 (SD) seconds

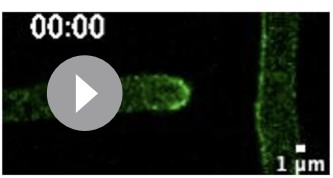

**Video 5.** Time course of GFP-SofT during a hyphal fusion event.

https://elifesciences.org/articles/83310/figures#video5

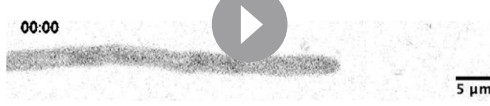

**Video 6.** Time course of MakB-mCherry during hyphal tip growth.

https://elifesciences.org/articles/83310/figures#video6

Co-localization of GFP-SofT (depicted in green) and MakB-mCherry (depicted in magenta) during hyphal tip growth.

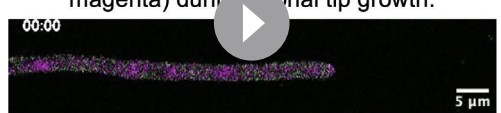

**Video 7.** Co-localization of GFP-SofT (depicted in green) and MakB-mCherry (depicted in magenta) during hyphal tip growth.

https://elifesciences.org/articles/83310/figures#video7

Co-localization of GFP-SofT (depicted in green) and MakB-mCherry (depicted in magenta) during a hyphal fusion event.

**Video 8.** Co-localization of GFP-SofT (depicted in green) and MakB-mCherry (depicted in magenta) during a hyphal fusion event.

https://elifesciences.org/articles/83310/figures#video8

($n$=105 in 11 hyphae) and 103±27 ($n$=106 in 10 hyphae), respectively (*Figure 3—figure supplement 2a, b*, *Video 9*).

A central assumption underlying the cell-to-cell dialogue in our model is that hyphae communicate by passing back and forth a shared signaling component, $Z$. Considering the central role of $Ca^{2+}$ in many types of cellular excitatory dynamics, and previous results that suggest that $Ca^{2+}$ is involved in the cell-to-cell dialogue (*Palma-Guerrero et al., 2013*; *Simonin et al., 2010*; *Fu et al., 2011*; *Fu et al., 2014*), we monitored intracellular $Ca^{2+}$ concentrations using the genetically encoded fluorescent reporter R-GECO. The fluorescent signals showed robust oscillations that were coordinated between two approaching hyphae (*Figure 4a and b*, *Video 10*; *Zhao et al., 2011*). The mean oscillation period of 104±33 (SD) seconds ($n$=160 in 24 hyphae) was comparable to the other markers during the cell-to-cell dialogue (*Figure 3c*, *Figure 3—figure supplement 2a,b*). Indeed, simultaneous visualization of R-GECO and GFP-ChsB showed synchronized oscillation with similar periods, indicating the anti-phasic oscillations of growth during the chemotropic interaction of the two hyphae (*Figure 4d and e*, *Video 11*). This phenomenon was coordinated in anti-phase in interacting hyphae. These results indicate that signaling and growth during the cell-to-cell dialogue are highly synchronized and possibly mediated by the uptake of $Ca^{2+}$.

To further test the role of $Ca^{2+}$, we depleted $Ca^{2+}$ from the media by adding the $Ca^{2+}$ chelating agent EGTA to LNA containing 1 µM $CaCl_2$. At an EGTA concentration of 5 mM, cell-fusion events were never observed after incubation for 16 up to 72 hr (*Figure 4—figure supplement 1a*). In addition, hyphal growth of *A. flagrans* was reduced, but germination of spores was unaffected. Fluorescence of R-GECO was not detectable in these $Ca^{2+}$-depleted conditions (*Figure 4—figure supplement 1b*). mCherry-ChsB still localized at hyphal tips, however, no obvious oscillating dynamic recruitment was observed (*Figure 4f*). mCherry-ChsB localized directly to the plasma membrane at the tip, without prior accumulation at the Spitzenkörper, confirming with previous reports the importance of $Ca^{2+}$ for well-regulated pulsatile secretion (*Kurian et al., 2022*). The localization of GFP-SofT to hyphal tips was abolished after the addition of 5 mM EGTA, indicating $Ca^{2+}$-dependent recruitment to the plasma membrane (*Figure 4g*). In line with these experimental results, reducing the external media concentration of signaling components in our simulations also abolished oscillations (*Figure 4—figure supplement 2*). On the other hand, external $Ca^{2+}$ concentrations of up to 100 mM did not influence hyphal fusion. Taken together, these results

Time course of Lifeact-GFP during a hyphal fusion event.

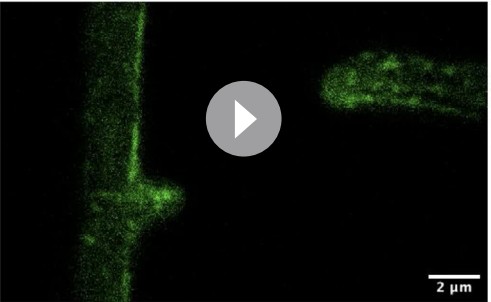

**Video 9.** Time course of Lifeact-GFP during a hyphal fusion event.

https://elifesciences.org/articles/83310/figures#video9

Time course of R-GECO during a hyphal fusion event.

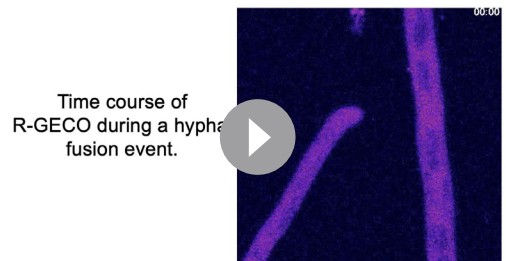

**Video 10.** Time course of R-GECO during a hyphal fusion event.

https://elifesciences.org/articles/83310/figures#video10

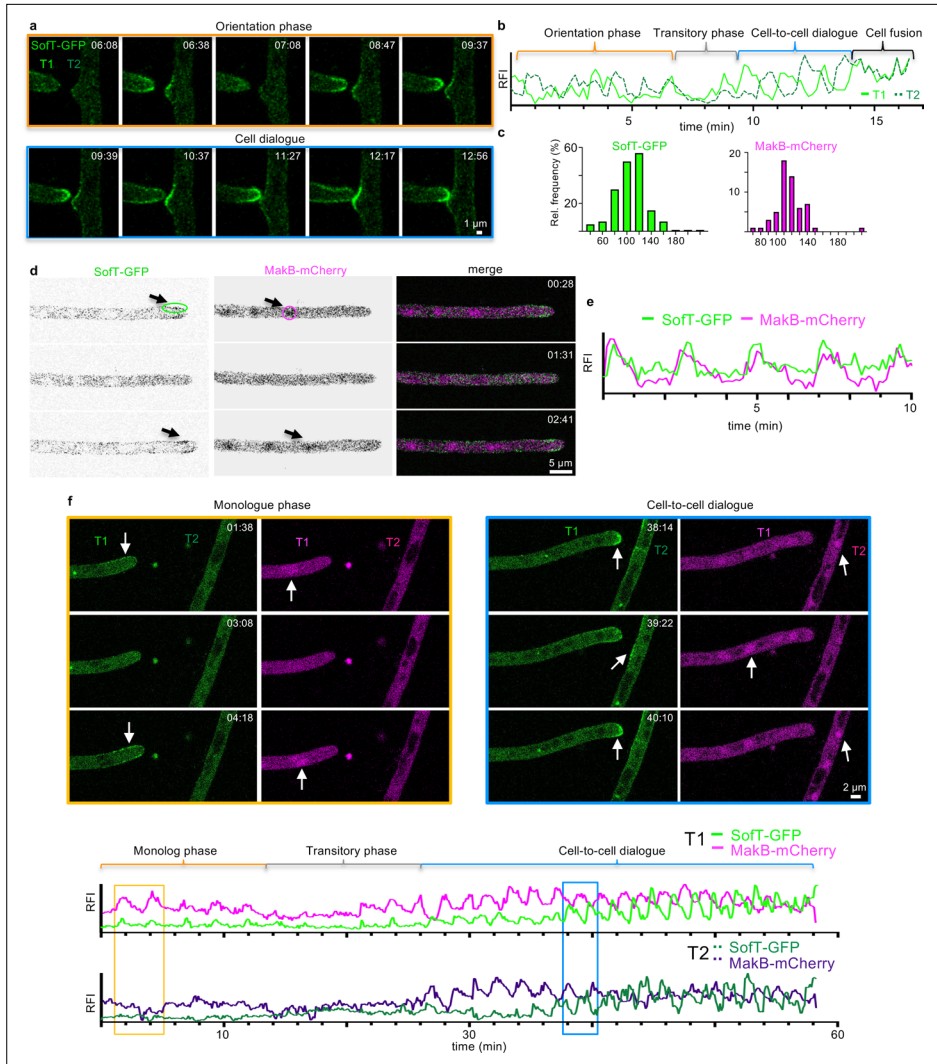

**Figure 3.** Decoupling of of SofT and MakB oscillations entering the cell-to-cell dialogue. (**a, b**) Time course of SofT-GFP during a hyphal fusion event. Selective micrographs of the time course are shown and are divided into an orientation phase (orange frame) and a cell-to-cell dialogue phase (blue frame). An area of 3×3 pixels at each hyphal apex was used to measure the fluorescent intensity depicted in (**b**) (y-axis, arbitrary units). T1=left tip; T2=right tip. (**c**) The interval between two peaks of SofT-GFP or MakB-mCherry at each hyphal tip during the cell dialogue was counted and the distribution is depicted as relative frequency (y-axis) over the time (in seconds). SofT-GFP *n*=178 in 18 hyphae. MakB-mCherry *n*=57 in 8 hyphae. (**d**) Co-localization of GFP-SofT (depicted in green) and MakB-mCherry (depicted in magenta) during hyphal tip growth. Arrows indicate the accumulation of the proteins at the hyphal tips. The circles indicate an area of 15×4 (SofT) and 10×10 (MakB) pixels which were used to measure the fluorescent intensity over time depicted in (**e**) of GFP-SofT at the upper edge of the plasma membrane or of MakB-mCherry in the nucleus. Single channels are depicted as inverted grayscale. (**e**) Relative fluorescent intensity (RFI, y-axis, arbitrary units) at the hyphal tips of (**d**) was measured over the time course (x-axis in minutes). (**f**) Time course of SofT-GFP and MakB-mCherry during a hyphal fusion event. Selective micrographs of the time course are shown and are divided into a monologue phase (orange frame) and cell-to-cell dialogue phase (blue frame). T1=left tip; T2=right hypha. Arrows indicate the accumulation of the proteins in the hyphae. An area of 21×21 pixels was used to measure the fluorescent intensity of MakB-mCherry, an area of 10×6 pixels was used to measure the fluorescent intensity of SofT-GFP depicted in the lower graphs (y-axis, arbitrary units). Each graph shows the fluorescence of one hypha.

The online version of this article includes the following figure supplement(s) for figure 3:

**Figure supplement 1.** Oscillatory recruitment of MakB-mCherry during hyphal growth of *A. flagrans*.

**Figure supplement 2.** The oscillating extension of hyphae is synchronized during hyphal fusion in *A.flagrans*.

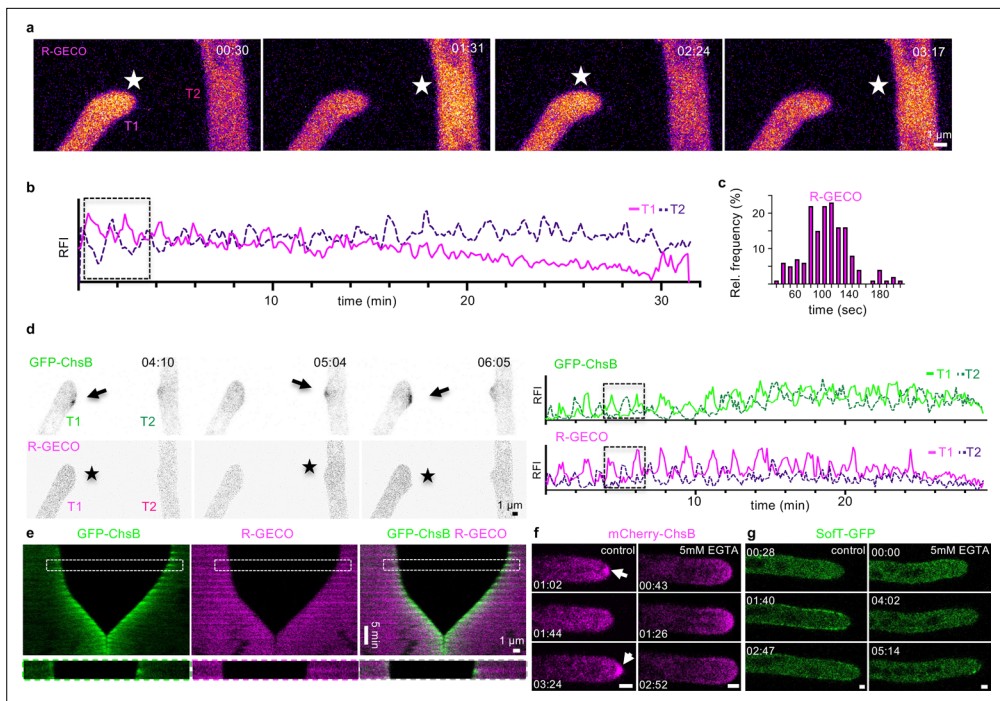

**Figure 4.** The stepwise extension of hyphae is coordinated during the cell-to-cell dialogue stage of hyphal fusion. Maximum-intensity projection of a time course of R-GECO during a hyphal fusion event. Stars indicate the fluorescent excitation of R-GECO in the presence of $Ca^{2+}$ inside the hyphae. The changes in fluorescent intensity are color-coded as Fire LUT using Fiji, depicting high pixel values as white and yellow color, and low pixel values as blue and magenta. The relative fluorescent intensity (RFI) (y-axis, arbitrary units) was measured at each hyphal tip over the time course (x-axis in minutes). The RFI of an area of 12×12 pixels was measured at each hyphal tip to generate the graph. The boxed area depicts the selected micrographs of the time course. (**c**) The interval between two peaks of R-GECO at each hyphal tip during the cell-to-cell dialogue was counted and the distribution is depicted as relative frequency (y-axis) over the time (in seconds). *n*=160 in 24 hyphae. (**d**) Maximum-intensity projection of a time course of GFP-ChsB and R-GECO during a hyphal fusion event. Single channels are depicted as inverted grayscale. Arrows indicate the localization of GFP-ChsB at hyphal tips. Stars indicate the fluorescent excitation of R-GECO in the presence of $Ca^{2+}$ inside the hyphae. The RFI (y-axis) was measured of an area of 6×6 pixels at each hyphal apex over the time course (x-axis in minutes, y-axis, arbitrary units). The boxed area of the selected micrographs represents the enlarged area in (**e**). (**e**) A kymograph was created of the time course in (**d**) drawing a line (pixel width 5) along the growth axis of both hyphae. The boxed area is enlarged and displays one cycle of the cell-to-cell dialogue. (**f**) Compared to the control (low-nutrient agar [LNA] containing 1 μM CaCl₂), mCherry-ChsB did not accumulate at the Spitzenkörper on LNA containing 5 mM EGTA. (**g**) Compared to the control, GFP-SofT did not accumulate at hyphal tips on LNA containing 5 mM EGTA. Scale bars in (**f, g**) depict 1 μm.

The online version of this article includes the following figure supplement(s) for figure 4:

**Figure supplement 1.** Extracellular $Ca^{2+}$ is necessary for hyphal fusion and pulse-like exocytosis.

**Figure supplement 2.** Sufficient concentration of $Ca^{2+}$ is necessary to produce oscillations.

indicate that extracellular $Ca^{2+}$ is important in the signaling mechanism that underlies the monologue implicated in pulsatile cell extension as well as the synchronization of oscillations into a cell-to-cell dialogue (*Figure 5*). An estimate calculation indicates that particles of diameter of approximately 3 nm or less can traverse the distance at which two hyphae synchronize at a sufficiently short time to support synchronization (for details, see Materials and methods). This diameter would include $Ca^{2+}$ ions as well as $Ca^{2+}$-binding proteins, but not secretory vesicles as the component that is exchanged between the hyphae.

During our study we identified and localized the SofT orthologue in *A. nidulans*. While we observed punctate localization throughout the hyphae resembling that seen in *A. flagrans*, we did not observe any tip oscillations in *A. nidulans* during various cultivation conditions (***Figure 5—figure supplement***

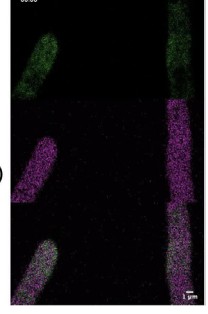

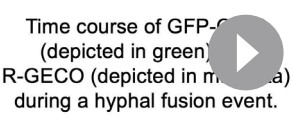

**Video 11.** Time course of GFP-ChsB (depicted in green) and R-GECO (depicted in magenta) during a hyphal fusion event.

https://elifesciences.org/articles/83310/figures#video11

1). Furthermore, we did not observe any hyphal fusion events. It is worth noting that *Aspergillus* species are known to rarely undergo vegetative cell fusion under laboratory conditions, compared to other fungi (*Macdonald et al., 2019*). This raises interesting questions for future research, such as investigating the presence of similar signaling dynamics during hyphal growth and sexual development in other fungi and investigating the factors that induce or regulate these dynamics in species like *A. nidulans*.

In order to further investigate the role of extracellular $Ca^{2+}$ during fungal communication, we concentrated on proteins which potentially play a role in incorporating the $Ca^{2+}$ signal into hyphal growth and the cell dialogue. Initially, we focused on orthologues of the protein FIG1 (mating factor induced gene 1) which is part of the low-affinity calcium uptake system in *Saccharomyces cerevisiae* and is essential for mating (*Muller et al., 2003*). We identified the orthologues FigA and FigB in the genome of *A. flagrans*. *figA* single gene deletion resulted in a colony phenotype, however, both gene deletion mutants still performed cell fusions and trapped nematodes (*Figure 5—figure supplement 2*). We were unable to generate a double-deletion strain, suggesting at least one protein to be essential for growth.

## Discussion

Fungal hyphae communicate with each other before they fuse and alternate between signal sender and signal perceiver functions, or between talking and listening if compared to people. Interestingly, the same signal oscillations occur in hyphae without any other hyphae in the vicinity. We named this phenomenon 'monologue' and show that - upon appearance of a fusion partner - the monologue transits into a dialogue. In comparison to people's conversations this means that people in a dark room walk around and whisper constantly until they meet a conversation partner. Now the two monologues need a transition phase that talking alternates with listening. For two hyphae trying to start a dialogue, the model predicted such a transitory phase, which we were able to monitor experimentally. At the molecular level the phenomenon may be explained by an interference of the signaling molecules of two approaching hyphae. The threshold concentrations for the switch between sending and receiving could be reached too early to be processed by the downstream components and hence the machinery would be disturbed until both hyphae respond to the common signal concentrations between the hyphal tips.

The growing tip of filamentous fungi is an example of apical-growing cells and therefore our results may have an impact on the understanding of other highly polarized cells such as plant pollen tubes or root hairs, or axons and dendrites (*Damineli et al., 2022*). Interestingly, the Soft protein exhibits weak homology to mammalian proteins of the aczonin/piccolo family, which are involved in synaptic vesicle release regulated by $Ca^{2+}$ (*Goryachev et al., 2012*). We screened for PRO40 (Soft)-interacting proteins with calcium-dependent functions in the fungus *Sordaria macrospora* identifying HAM-10, an orthologue of Unc13/Munc13 in higher eukaryotes (*Teichert et al., 2014*). HamA, the *A. flagrans* orthologue, contains a calcium-dependent C2 domain similar to other fungal orthologues. Deletion of Δ*hamA* resulted in a cell fusion mutant phenotype, as observed in the Δ*ham-10* mutant in *N. crassa* (*Fu et al., 2011*; *Figure 5—figure supplement 3*). Interestingly, in the Δ*ham-10* mutant, neither Soft nor MAK-2 localized at the tip of germ tubes (*Fu et al., 2014*). We localized mCherry-ChsB in the Δ*hamA* mutant and observed oscillating recruitment of the protein to the hyphal tip (*Figure 5—figure supplement 3*), indicating that HamA could play a role at the interface of combining cell dialogue signaling and growth dynamics. RIM1 is another Munc13-interacting protein, which plays a role in linking vesicle fusion and calcium influx. In *S. macrospora*, the orthologue of RIM1 was identified as PRO40-interacting protein (*Teichert et al., 2014*). In *N. crassa* the orthologous Syt1 is involved in cell fusion, but not essential (*Palma-Guerrero et al., 2013*). The orthologue RimA in *A. flagrans* localized

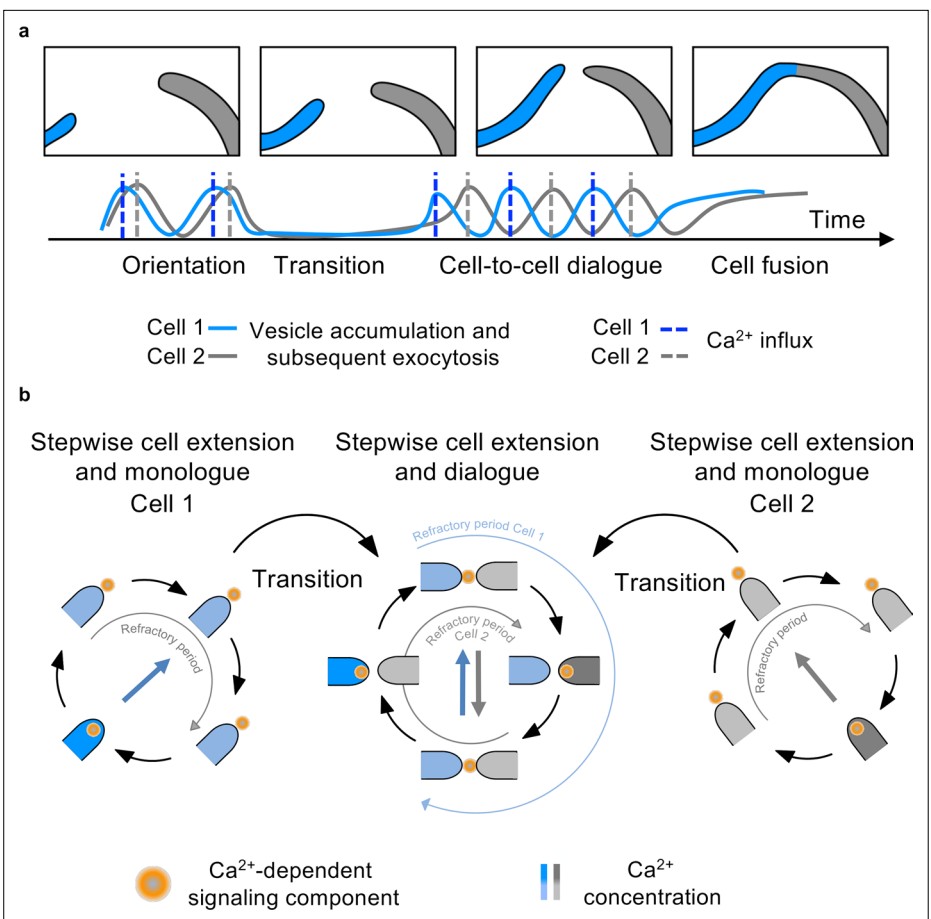

**Figure 5.** Scheme of the cell-to-cell dialogue. (**a**) Cell fusion of two hyphal cells is mediated by the synchronization of oscillatory growth. Vesicles needed for hyphal growth and communication accumulate at the hyphal tip and are released to the surroundings upon an influx of $Ca^{2+}$. If two cells are in close proximity, the uncoordinated, stepwise growth shifts after transitory entrainment to a synchronized anti-phasic cell dialogue and subsequent cell fusion. (**b**) During stepwise cell extension, a $Ca^{2+}$-dependent signaling component is released with a refractory period to prevent self-stimulation. Entrainment of two cells in close proximity initiates a cell dialogue mediated by a so far unknown $Ca^{2+}$-dependent signaling component. Refractory periods after secretion prevent self-stimulation.

The online version of this article includes the following figure supplement(s) for figure 5:

**Figure supplement 1.** Localization of Soft-GFP in *A. nidulans*.

**Figure supplement 2.** Gene deletion of *figA* and *figB* in *A. flagrans*.

**Figure supplement 3.** Gene deletion of hamA leads to loss of hyphal fusion.

---

at the hyphal tip without noticeable oscillations during our time series (*Figure 5—figure supplement 3*, *Video 12*). Calmodulin and a calcium/calmodulin-dependent protein kinase were identified as PRO40 interaction partners (*Teichert et al., 2014*), indicating intriguing similarities between the rapid secretion of the cell-to-cell dialogue in fungi and synaptic vesicle release in neurons. In the future, it will be interesting to study how other fusion-related proteins might be involved in the translation of the increase in intracellular $Ca^{2+}$ concentrations and how this change relates to the secretion of a yet to be identified chemoattractive signal molecule.

Time course of mCherry-RimA during hyphal tip growth.

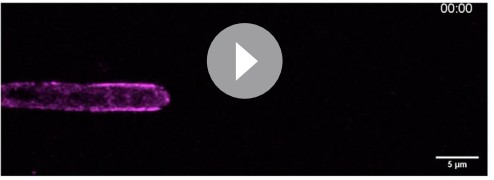

**Video 12.** Time course of mCherry-RimA during hyphal tip growth.

https://elifesciences.org/articles/83310/figures#video12

**Table 1.** *A. flagrans* and *A. nidulans* strains used in this study.

| Strain number | Genotype | Origin |
|---|---|---|
| CBS 349.94 | Wild type | *Youssar et al., 2019* |
| sVW16 | *tubA(p)::lifeact::GFP::gluC(t); trpC(p)::hph::trpC(t)* | *Wernet et al., 2022* |
| sVW01 | *sofT::hph (Δsoft)* | *Youssar et al., 2019* |
| sVW20 | *ΔsofT; sofT(p)::sofT::gluC(t); neo; makB(p)::makB::makB(t); ntc* | *Haj Hammadeh et al., 2022* |
| sVW25 | *tubA(p)::lifeact::GFP::gluC(t); trpC(p)::hph::trpC(t)* | *Wernet et al., 2022* |
| sVW29 | *sofT(p)::sofT::GFP::sofT(t); trpC(p)::hph::trpC(t)* | This study |
| sVW40 | *soft(p)::sofT (Δ1–504 AA N-terminus) gpdA(p)::neo::trpC(t)* | This study |
| sVW41 | *soft(p)::sofT (Δ534–1213 AA C-terminus) gpdA(p)::neo::trpC(t)* | This study |
| sVW43 | *soft(p)::sofT (Δ1–540 AA N-terminus, no WW domain) gpdA(p)::neo::trpC(t)* | This study |
| sVW51 | *tubA(p)::mCherry::chsB::chsB(t); gpdA(p)::neo::trpC(t)* | This study |
| sVW52 | *sofT(p)::GFP::sofT; tubA(p)::mCherry::chsB::chsB(t);trpC(p)::hph::trpC(t); gpdA(p)::neo::trpC(t)* | This study |
| sVW65 | *h2b(p)::r-GECO::gluC(t); gpdA(p)::neo::trpC(t)* | This study |
| sVW69 | *h2b(p)::r-GECO::gluC(t);tubA(p)::GFP::chsB::chsB(t); gpdA(p)::neo::trpC(t); trpC(p)::hph::trpC(t)* | This study |
| sVW75 | *sofT::hph (Δsoft); tubA(p)::mCherry::chsB::chsB(t); gpdA(p)::neo::trpC(t); trpC(p)::hph::trpC(t)* | This study |
| sVW76 | *figA::hph (ΔfigA)* | This study |
| sVW77 | *figB::hph (ΔfigB)* | This study |
| sVW78 | *tubA(p)::mCherry::rimA::chsB(t); gpdA(p)::neo::trpC(t)* | This study |
| sVW44 | *hamA::hph (ΔhamA), trpC(p)::hph::trpC(t)* | This study |
| sVW80 | sVW51,<br>*hamA::hph (ΔhamA), trpC(p)::hph::trpC(t)* | This study |
| RPA 33 | *yA1, riboB2, pyrG89, pyroA4, pabaA1, nkuA::argB+* | |
| VW-a01 | *yA1, riboB2, pyrG89, (AnSoft-GFP::Afpyro) pyroA4, pabaA1, nkuA::argB+* | This study |

## Materials and methods
### Strains and culture conditions

*A. flagrans* strains (all derived from CBS349.94) were cultivated at 28°C on PDA (2.4% potato dextrose broth and 1.5% agar, Carl Roth). All fungal strains used in the study are listed in *Table 1*. Protoplast transformation was performed as described in *Youssar et al., 2019*. Chemically competent *Escherichia coli* Top10 cells were used for plasmid cloning. *A. nidulans* strains were cultivated at 37°C on *Aspergillus* minimal media and transformed as described in *Szewczyk et al., 2006*.

### Plasmid and strain construction

All plasmids and primers used in the study are listed in *Tables 2 and 3*. Chemically competent *E. coli* Top10 cells were used for plasmid cloning. *A. flagrans* SofT was tagged with GFP at the C-terminus and expressed under the native promoter at the *sofT* locus. A 1 kb region from the 3'-end of the sofT *open reading frame* (orf) and 0.6 kb of the terminator region were amplified by PCR from *A. flagrans* genomic DNA. A *GFP::hph* cassette was amplified from pNH57 and all three fragments were subsequently inserted into the linearized plasmid backbone pJET1.2 using Gibson assembly, resulting in plasmid pVW106. *A. flagrans* ChsB was tagged with either mCherry or GFP at the N-terminus and expressed under the alpha-tubulin *tubA* promoter. The *chsB* orf and the 0.9 kb 3' region were amplified by PCR from *A. flagrans* gDNA and inserted into the plasmid backbone pVW92 using Gibson assembly, resulting in the *tubA-mCherry-chsB* plasmid pVW118. For a GFP-tagged ChsB variant, the *mCherry* cassette of pVW118 was replaced by *GFP* using Gibson assembly, resulting in pVW132.

R-GECO was expressed under the histone *h2b* promoter. The R-GECO sequence was amplified from gDNA of SNT162 (*Takeshita et al., 2017*) and cloned into plasmid pVW92, resulting in pVW120.

**Table 2.** Plasmids used in this study.

| Name | Description/genotype | Reference |
|---|---|---|
| pVW57 | Plasmid backbone containing *trpC(p)::hph::trpC(t)* | *Wernet et al., 2022* |
| pVW92 | Plasmid backbone containing *gpdA(p)::neo::trpC(t)* and *tubA(p)* | *Wernet et al., 2021* |
| pNH57 | Plasmid containing *GFP::hph* | Nicole Wernet (Karlsruhe) |
| pVW106 | *sofT*(p)::*sofT*::GFP::*sofT*(t); *trpC(p)::hph::trpC(t)* | This study |
| pVW118 | *tubA(p)::mCherry::chsB::chsB(t);gpdA(p)::neo::trpC(t)* | This study |
| pVW132 | *tubA(p)::GFP::chsB::chsB(t); gpdA(p)::neo::trpC(t)* | This study |
| pVW120 | *tubA(p)::r-GECO::gluC(t); gpdA(p)::neo::trpC(t)* | This study |
| pVW125 | *h2b(p)::r-GECO::gluC(t); trpC(p)::hph::trpC(t)* | This study |
| pVW131 | *h2b(p)::r-GECO::trpC(t); trpC(p)::hph::trpC(t)* | This study |
| pVW132 | *figA::hph (ΔfigA)* | This study |
| pVW133 | *figB::hph (ΔfigB)* | This study |
| pVW134 | *tubA(p)::mCherry::rimA::chsB(t); gpdA(p)::neo::trpC(t)* | This study |
| pVW113 | *hamA::hph (ΔhamA)* | This study |
| pVW-a68 | *AnSoft-GFP::Afpyro* | This study |
| pVW108 | *soft(p)::sofT (Δ1–504 AA N-terminus) gpdA(p)::neo::trpC(t)* | This study |
| pVW109 | *soft(p)::sofT (Δ534–1213 AA C-terminus) gpdA(p)::neo::trpC(t)* | This study |
| pVW112 | *soft(p)::sofT (Δ1–540 AA N-terminus, no WW domain) gpdA(p)::neo::trpC(t)* | This study |

Subsequently, the *tubA* promoter was exchanged by the 1 kb sequence of the *h2b* promoter using Gibson assembly, resulting in plasmid pVW125. For co-localization experiments, the *h2b*(p)-*R-GECO*-fragment was cloned into pVW57, resulting in pVW131.

*figA*, *figB*, and *hamA* were deleted by homologous recombination. One kb flanks homologous to the 5' and 3' regions of the gene of interest were amplified by PCR with 25 bp overhangs homologous to either the hygromycin-B (*hph*) or geneticin sulfate (G418, *neo*) resistance cassette as well as to the pJET1.2 plasmid backbone. Verification of homologous recombination was performed as described in *Wernet et al., 2022*.

*A. flagrans* RimA was tagged with mCherry at the N-terminus and expressed under the *tubA* promoter. The *rimA orf* and 3' region were amplified by PCR from *A. flagrans* gDNA and assembled into the plasmid backbone pVW82 using Gibson assembly.

Fragments of the *sofT orf* were amplified by PCR from *A. flagrans* gDNA and assembled into the plasmid backbone pJM16 using Gibson assembly for complementing the *ΔsofT* mutant with truncated versions of the protein. Each construct was expressed under the *sofT* promoter.

*A. nidulans* Soft was tagged with GFP at its endogenous locus with the endogenous promoter. Fragments were amplified by PCR from *A. nidulans* gDNA and assembled into the Blue Heron Biotechnology pUC vector.

## Microscopy

For microscopy, fungal strains were inoculated on thin LNA (1 g/l KCl, 0.2 g MgSO4 - 7$H_2$O, 0.4 mg MnSO$_4$ - 4$H_2$O, 0.88 mg ZnSO$_4$ - 7$H_2$O, 3 mg FeCl$_3$ - 6$H_2$O, 15 g agar, pH 5.5) slides supplemented with 1 µM CaCl$_2$. For calcium chelating experiments, EGTA (stock solution 0.5 M) was added at a final concentration of 5 mM. Around 1×10$^4$ *A. flagrans* spores were incubated on a 2×2 cm agar pad at 28°C in darkness for 12–72 hr.

Live cell imaging of *A. flagrans* was performed using a confocal microscope (LSM900, Carl Zeiss) with a 63× NA 1.4 oil objective lens (DIC M27). Time series were acquired with a gallium arsenide phosphide photomultiplier tube (GaAsP-PMT) detector, 488 or 561 nm excitation lasers were used. Live cell imaging of *A. nidulans* was performed using a Nikon Ti2 microscope mounted with a Yokogawa

**Table 3.** Oligonucleotides used in this study.

| Name | Sequence (5' to 3') | Description |
|---|---|---|
| chsB_fwd | gaatggatgaactctacaaa atggcacagcaaggaggtt | mCherry-ChsB |
| chsB_rev | aggagatcttctagaaagatgatggggcgttaaggtttc | mCherry-ChsB |
| pJet_fwd | atctttctagaagatctcctacaatattc | mCherry-ChsB |
| mCherry_rev | tttgtagagttcatccattccac | mCherry-ChsB |
| tubP_rev | gatgaattatatttcgtcaagaag | GFP-ChsB; tubA(p)::r-geco |
| chsb_fwd | atggcacagcaaggaggtt | GFP-ChsB |
| gfp_chsb_fwd | ttgacgaaatataattcatcatggtttccaagggtgagg | GFP-ChsB |
| gfp_chsb_rev | taacctccttgctgtgccatagcggccgctttgtaaagtt | GFP-ChsB |
| R_geco_pOL_fwd | ttgacgaaatataattcatcatggtcgactcatcacgtc | R-GECO (*tubA*(p)) |
| r-geco_tOL_rev | atacatcttatctacatacgctacttcgctgtcatcatttg | R-GECO (*tubA*(p)) |
| tgluC_gOL_fwd | aaatgatgacagcgaagtagcgtatgtagataagatgtatgattag | R-GECO (*tubA*(p)) |
| tgluC_geco_rev | aggagatcttctagaaagatatcttgttgggggggaaggg | R-GECO (*tubA*(p)) |
| h2b_p_trpCOL_fwd | ctttccctaaactcccccccaggagaagaaaggagcaaaatc | R-GECO (*h2b*(p)) |
| h2b_p_gecoOL_rev | cgacgtgatgagtcgaccattttgaaatttgtttttgtttgggtag | R-GECO (*h2b*(p)) |
| trpC_rev | tggggggagtttagggaaag | R-GECO (*h2b*(p)) |
| r_geco_fwd | atggtcgactcatcacgtc | R-GECO (*h2b*(p)) |
| soft_gfp_locus_fwd | ctcgagtttttcagcaagattaccgtcctcagtacaacatg | SofT-GFP |
| soft_gfp_locus_rev | acctcacccttggaaaccatatacccatactcgcatctgg | SofT-GFP |
| soft_GFPcas_fwd | ccagatgcgagtatgggtatatggtttccaagggtgagg | SofT-GFP |
| soft_GFPcas_rev | caaccgcccggacgaatcattggggggagtttagggaaag | SofT-GFP |
| soft_Term_fwd | ctttccctaaactcccccccaatgattcgtccgggcggtt | SofT-GFP |
| softterm_gfp_rev | attgtaggagatcttctagaaagattgggacgagtgggatttaaaatgga | SofT-GFP |
| figA_lb_fwd | ctcgagtttttcagcaagatTGTCGCTTGGGCTTGATAG | Deletion *figA* |
| figA_lb_rev | CCTCCACTAGCATTACACTTATCTGCTAACGTAACTAGACG | Deletion *figA* |
| figA_hph_fwd | GTCTAGTTACGTTAGCAGATAAGTGTAATGCTAGTGGAGG | Deletion *figA* |
| figA_hph_rev | CTAACAGGCCTATCGGAGTTTGGGGGGGAGTTTAGGGAAAG | Deletion *figA* |
| figA_rb_fwd | CTTTCCCTAAACTCCCCCCAAACTCCGATAGGCCTGTTAG | Deletion *figA* |
| figA_rb_rev | aggagatcttctagaaagatCGGAGGTCGTCAAGAAGC | Deletion *figA* |
| figA_up_fwd | AGGAAGACCGATTACGAAAC | Deletion *figA* |
| figA_down_rev | CGATATACGATCCGAAGGTC | Deletion *figA* |
| figA_lb_g418_rev | AATGCAATGTAATAGATACCATCTGCTAACGTAACTAGACG | Deletion *figA* |
| figA_g418_fwd | GTCTAGTTACGTTAGCAGATGGTATCTATTACATTGCATTGCG | Deletion *figA* |
| figB_lb_fwd | ctcgagtttttcagcaagatGGCGAAGAGACTGGATTTATC | Deletion *figB* |
| figB_lb_rev | CCTCCACTAGCATTACACTTTTTGAAGTTTTGCTGATGATGTGAG | Deletion *figB* |
| figB_hph_fwd | ATCATCAGCAAAACTTCAAAAAGTGTAATGCTAGTGGAGGT | Deletion *figB* |
| figB_hph_rev | TTGTTCAGCTTTTTTCCCATTGGGGGGGAGTTTAGGGAAAG | Deletion *figB* |
| figB_rb_fwd | CTTTCCCTAAACTCCCCCCAATGGGAAAAAAGCTGAACAAAAAAAATC | Deletion *figB* |
| figB_rb_rev | aggagatcttctagaaagatCGCCTGTGTAACGGCTTTTG | Deletion *figB* |

*Table 3 continued on next page*

*Table 3 continued*

| Name | Sequence (5' to 3') | Description |
|---|---|---|
| figB_up_fwd | AGAGCCGCATGGTTTATTTAG | Deletion *figB* |
| figB_down_rev | AGCACAGAGTAACCTGGAC | Deletion *figB* |
| rimA_orf_fwd | GAATGGATGAACTCTACAAAATGGAAACCCCAGCTCCAG | mCherry-RimA |
| rimA_rb_rev | AGGAGATCTTCTAGAAAGATCGTTCTATGCCTGAAATCGG | mCherry-RimA |
| ham10_lb_fwd | ctcgagtttttcagcaagatGGCGGATATCAATCTTATCTTG | Deletion *hamA* |
| ham10_lb_rev | CCTCCACTAGCATTACACTTGGTGACCGAAATCGCCTTAT | Deletion *hamA* |
| ham10_hph_fwd | ATAAGGCGATTTCGGTCACCAAGTGTAATGCTAGTGGAGG | Deletion *hamA* |
| ham10_hph_rev | CACAAGGATGGCTTCCCATTTGGGGGGAGTTTAGGGAAAG | Deletion *hamA* |
| ham10_rb_fwd | CTTTCCCTAAACTCCCCCCAAATGGGAAGCCATCCTTGTG | Deletion *hamA* |
| ham10_rb_rev | aggagatcttctagaaagatCGTTCCGATTTACTCGTCG | Deletion *hamA* |
| ham10_up_fwd | GCCGAGACTACTAGCTAGG | Deletion *hamA* |
| ham10_down_rev | CTATATCGTTGGTTCGAGGG | Deletion *hamA* |
| soft_Nterm_tgluCOL_rev | ATACATCTTATCTACATACGTTAAATACCGGTCTCCGGTGC | N-terminal SofT truncation |
| soft_ntermnoWW_tgluCol_rev | ATACATCTTATCTACATACGTTATGGAAGAGGTGGGGGAGA | N-terminal SofT truncation, no WW domain |
| softProm_trpCOL_fwd | CTTTCCCTAAACTCCCCCCACAGAGTTCGAATAGCGTTGC | C-terminal SofT truncation |
| softProm_CtermOL_rev | ATACCGGTCTCCGGTGCCATTGTGGAGACGAAGGCAAAG | C-terminal SofT truncation |
| softCterm_fwd | CCTTTGCCTTCGTCTCCACAATGGCACCGGAGACCGGTATT | C-terminal SofT truncation |
| softCterm_tgluC_rev | ATACATCTTATCTACATACGTTAATACCCATACTCGCATCTG | C-terminal SofT truncation |
| VW230 | TTGTAAAACGACGGCCAGTGACACCAGAAATCTCTCCGAATG | Soft-GFP, *A. nidulans* |
| VW231 | GTGAAAAGTTCTTCTCCTTTCTTCCCATGCTCTAAACTCG | Soft-GFP, *A. nidulans* |
| VW232 | CGAGTTTAGAGCATGGGAAGAAAGGAGAAGAACTTTTCACTGG | Soft-GFP, *A. nidulans* |
| VW233 | TTAAAAAGACTCGGCATCAAgcgagtgtctacataatgaagg | Soft-GFP, *A. nidulans* |
| VW234 | ttcattatgtagacactcgcTTGATGCCGAGTCTTTTTAATGT | Soft-GFP, *A. nidulans* |
| VW235 | GACCATGATTACGCCAagctGGGTGACGGATTATTACCTCT | Soft-GFP, *A. nidulans* |

W1 spinning-disk confocal scan head and two Prime95B cameras. Images were acquired with the NIS Elements Advanced Research software (Nikon) with the 488 nm laser.

Image processing and analysis were performed in FIJI (*Schindelin et al., 2012*) and ZEN Blue. Datasets were analyzed in GraphPad Prism. Kymographs were generated with the FIJI Multi Kymograph tool using line widths 5. The dynamics of the fluorescent intensity over time were measured in each frame with a circular selection at the respective hyphal tip, specified in each figure legend. The interval between two accumulating peaks at hyphal tips was counted in ZEN Blue.

All raw images and data files resulting from the analyses are available at Zenodo (https://zenodo.org/record/6830734#.Ys_KmS2w1TY).

To quantify SofT-GFP at hyphal tips, spores were incubated on a 2×2 cm agar pads at 28°C for 24 hr and the fluorescence of 50 tips was counted three times.

## Mathematical model

Our model is based on the previous model of the dialogue-like communication between two Neurospora cells growing toward each other (**Goryachev et al., 2012**). The original model consists of eight ordinary differential equations, four representing each cell. Those equations represent an excitatory system that coordinates the docking and release of vesicles with chemoattractants. We extended this model by explicitly modeling the dynamics of the chemoattractants in the extracellular space at the tips of the individual cells. This is modeled by two coupled differential equations that represent the concentration of the signaling molecules in the proximity of the corresponding cells. Those two equations are linked with a coupling whose magnitude represents the coupling strength. The model was implemented in the Julia Programming Language as a set of stochastic differential equations and simulated using the Euler-Maruyama method with integration step $dt$ = 0.001. The complete set of equations and parameter values can be found in the Supplementary Information. The simulation script is also available at https://github.com/vkumpost/cell-dialog (copy archived at **Kumpošt, 2022**).

The model was implemented in the form of stochastic differential equations as

$$
\dot{U} = \begin{bmatrix} \dot{A}_1 \\ \dot{I}_1 \\ \dot{X}_1 \\ \dot{Y}_1 \\ \dot{Z}_1 \\ \dot{A}_2 \\ \dot{I}_2 \\ \dot{X}_2 \\ \dot{Y}_2 \\ \dot{Z}_2 \end{bmatrix} = \tau F(U) + \sqrt{\tau} \frac{1}{\sqrt{\Omega}} G(U)
$$

where $U$ is a vector of state variables that represent activator ($A$), inhibitor ($I$), free vesicle ($X$), docked vesicle ($Y$), and concentration of the extracellular signaling molecule at the tip of the cell ($Z$). The index (1, 2) indicates the specific cell. $\tau$ is a time-scaling constant to adjust the time scale without the loss of dynamics. $\Omega$ is the system size and controls the level of noise in the system. The drift function ($F$) reads

$$
F(U) = \begin{bmatrix} a_0 - \alpha A_1 + \beta A_1^2 - A_1^3 - A_1 I_1 + A_1 Z_1 \\ \varepsilon \left( i_0 + \gamma \frac{A_1^2}{A_1^2 + k} - I_1 \right) \\ x_0 - k_1 A_1^2 X_1 \\ k_1 A_1^2 X_1 - k_2 \frac{1}{A_1^3 + 1} Y_1 \\ -k_{Cext} \left( Z_1 - C_{media} \right) + k_2 \frac{1}{A_1^3 + 1} Y_1 - A_1 Z_1 + k_z e^{-d} Z_2 - k_z e^{-d} Z_1 \\ a_0 - \alpha A_2 + \beta A_2^2 - A_2^3 - A_2 I_2 + A_2 Z_2 \\ \varepsilon \left( i_0 + \gamma \frac{A_2^2}{A_2^2 + k} - I_2 \right) \\ x_0 - k_1 A_2^2 X_2 \\ k_1 A_2^2 X_2 - k_2 \frac{1}{A_2^3 + 1} Y_2 \\ -k_{Cext} \left( Z_2 - C_{media} \right) + k_2 \frac{1}{A_2^3 + 1} Y_2 - A_2 Z_2 + k_z e^{-d} Z_1 - k_z e^{-d} Z_2 \end{bmatrix}
$$

The noise function ($G$) is

$$G(U) = \begin{bmatrix} \sqrt{a_0}\xi_{1,1} + \sqrt{\alpha A_1}\xi_{1,2} + \sqrt{\beta A_1^2}\xi_{1,3} + \sqrt{A_1^3}\xi_{1,4} + \sqrt{A_1 I_1}\xi_{1,5} + \sqrt{A_1 Z_1}\xi_{1,6} \\ \sqrt{\varepsilon i_0}\xi_{1,7} + \sqrt{\varepsilon \gamma \frac{A_1^2}{A_1^2+k}}\xi_{1,8} + \sqrt{\varepsilon I_1}\xi_{1,9} \\ \sqrt{x_0}\xi_{1,10} + \sqrt{k_1 A_1^2 X_1}\xi_{1,11} \\ \sqrt{k_1 A_1^2 X_1}\xi_{1,12} + \sqrt{k_2 \frac{1}{A_1^3+1}Y_1}\xi_{1,13} \\ \sqrt{k_{Cext}Z_1}\xi_{1,14} + \sqrt{k_{Cext}C_{media}}\xi_{1,15} + \sqrt{k_2 \frac{1}{A_1^3+1}Y_1}\xi_{1,16} + \sqrt{A_1 Z_1}\xi_{1,17} + \sqrt{k_z e^{-d}Z_2}\xi_{1,18} + \sqrt{k_z e^{-d}Z_1}\xi_{1,19} \\ \sqrt{a_0}\xi_{2,1} + \sqrt{\alpha A_2}\xi_{2,2} + \sqrt{\beta A_2^2}\xi_{2,3} + \sqrt{A_2^3}\xi_{2,4} + \sqrt{A_2 I_2}\xi_{2,5} + \sqrt{A_2 Z_2}\xi_{2,6} \\ \sqrt{\varepsilon i_0}\xi_{2,7} + \sqrt{\varepsilon \gamma \frac{A_2^2}{A_2^2+k}}\xi_{2,8} + \sqrt{\varepsilon I_2}\xi_{2,9} \\ \sqrt{x_0}\xi_{2,10} + \sqrt{k_1 A_2^2 X_2}\xi_{2,11} \\ \sqrt{k_1 A_2^2 X_2}\xi_{2,12} + \sqrt{k_2 \frac{1}{A_2^3+1}Y_2}\xi_{2,13} \\ \sqrt{k_{Cext}Z_2}\xi_{2,14} + \sqrt{k_{Cext}C_{media}}\xi_{2,15} + \sqrt{k_2 \frac{1}{A_2^3+1}Y_2}\xi_{2,16} + \sqrt{A_2 Z_2}\xi_{2,17} + \sqrt{k_z e^{-d}Z_1}\xi_{2,18} + \sqrt{k_z e^{-d}Z_2}\xi_{2,19} \end{bmatrix}$$

where $\xi$ are independent Wiener processes. The results presented in the manuscript were obtained for parameter values $\tau$=8.48, $\varepsilon$=0.55, $\alpha$=12.4, $\beta$=8.05, $\gamma$=8, $k$=6, $a_o$=5.6, $i_o$=0.1, $x_o$=1.0, $k_1$=0.1, $k_2$=1.0, $k_z$ = 10.0, $k_{cext}$ = 5, $C_{media}$ = 1, $d$=10 (long distance), $d$=0 (short distance), $\Omega$=1000. The initial conditions for the simulations were set to $U(t=0)$ = [0.7065, 0.7145, 0, 0, 1.0, 0.7065, 0.7145, 0, 0, 1.0]. To minimize the effect of the initial conditions, the model was run for 200 min before any other analysis was performed.

## Particle size estimation for the signaling component

Assuming Brownian diffusion in three dimensions, the mean square distance ($L^2$) traveled by a particle is

$$L^2 = 6Dt,$$

where $D$ is the diffusion constant and $t$ the time for which the particle is followed. Inserting for the diffusion coefficient on the basis of the Stokes-Einstein relationship

$$D = \frac{k_B T}{6\pi \eta r}$$

for a particle of radius $r$, with the Boltzmann constant $k_B$, the water temperature $T$=298.15 K, the dynamic viscosity $\eta = 0.89 \, \mathrm{mPa \cdot s}$ of water at 25°C (*The International Association for the Properties of Water and Steam, 2008*), we obtain

$$r \approx 1.5 \cdot 10^{-18} \frac{m^2}{s} \frac{t}{L^2}.$$

Inserting as approximate values a typical distance at which two hyphae enter into asynchronous dialogue ($L \approx 10 \, \mu\mathrm{m}$) and an upper bound estimate of the time that is allowed to travel from one hypha to another and sustain oscillations with a period of $\approx 60 \, \mathrm{s}$ ($t \approx 10 \, \mathrm{s}$), we can calculate an upper bound on the particle radius: $r \lesssim 1.5 \, \mathrm{nm}$. This upper bound implies a maximal particle diameter of $\approx 3 \, \mathrm{nm}$, so that both ionic species (typical diameters 1 nm) as well as typical proteins (typical diameters $1 - 3 \, \mathrm{nm}$) might take the role of the signaling component, while achieving a fast enough transfer between the two hyphae by conventional Brownian diffusion. Secretory vesicles (typical diameters $\geq 30 \, \mathrm{nm}$) exceed the upper bound, and would be unlikely to diffuse rapidly enough from one hypha to another.

## Materials availability statement

The manuscript includes a dedicated 'materials availability statement' providing transparent disclosure about availability of newly created materials including details on how materials can be accessed and describing any restrictions on access.

## Acknowledgements

We are thankful to the group of Prof. Dr. André Fleißner (University of Braunschweig) for fruitful discussions and Prof. Dr. Sylvia Erhardt (KIT) for the opportunity and the help to use the Zeiss LSM900 microscope. We thank Prof. Dr. Samara Reck-Peterson for help using the Nikon Ti2 microscope and providing *Aspergillus* strains. Valentin Wernet was funded by the German Federal Environmental Foundation (DBU). VK was funded by the Helmholtz Information and Data Science School for Health (HIDSS4Health). RM and LH were funded by the Helmholtz Program Natural, Artificial, and Cognitive Information Processing (NACIP).

## Additional information

### Funding

| Funder | Grant reference number | Author |
|---|---|---|
| Deutsche Forschungsgemeinschaft | Fi 459/26-1 | Reinhard Fischer |
| Deutsche Bundesstiftung Umwelt | 2018/552 | Valentin Wernet |
| Helmholtz Association | HIDSS4Health | Vojtech Kumpost Ralf Mikut Lennart Hilbert |

The funders had no role in study design, data collection and interpretation, or the decision to submit the work for publication.

### Author contributions

Valentin Wernet, Data curation, Formal analysis, Validation, Investigation, Visualization, Methodology, Writing - original draft; Marius Kriegler, Formal analysis, Investigation; Vojtech Kumpost, Software, Formal analysis, Investigation, Visualization, Methodology; Ralf Mikut, Supervision, Funding acquisition; Lennart Hilbert, Data curation, Software, Formal analysis, Supervision, Validation, Methodology; Reinhard Fischer, Conceptualization, Supervision, Funding acquisition, Validation, Project administration, Writing - review and editing

### Author ORCIDs

Valentin Wernet http://orcid.org/0000-0002-3747-6171
Marius Kriegler http://orcid.org/0009-0004-4395-4784
Lennart Hilbert http://orcid.org/0000-0003-4478-5607
Reinhard Fischer http://orcid.org/0000-0002-6704-2569

### Decision letter and Author response

Decision letter https://doi.org/10.7554/eLife.83310.sa1
Author response https://doi.org/10.7554/eLife.83310.sa2

## Additional files

### Supplementary files
• MDAR checklist

### Data availability

All raw images and data files resulting from the analyses are available at Zenodo (https://doi.org/10.5281/zenodo.6830734).The simulation script is also available at https://github.com/vkumpost/cell-dialog (copy archived at *Kumpošt, 2022*).

The following dataset was generated:

| Author(s) | Year | Dataset title | Dataset URL | Database and Identifier |
|---|---|---|---|---|
| Wernet V | 2022 | Data from: Establishment and synchronization of a cell-to-cell dialogue as prerequisite for cell fusion | https://doi.org/10.5281/zenodo.6830734 | Zenodo, 10.5281/zenodo.6830734 |

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
