## [Editor Report]

This important study combines convincing live cell imaging and mathematical modeling data to show how an emerging model fungus engages in an oscillatory chemical dialogue to prepare for cell-cell fusion. In the absence of a fusion partner, fungal hyphae undergo signal oscillations that are in phase with their growth oscillations; following detection of a fusion partner, the oscillation frequencies slow down, and a transition to an anti-phasic synchronization of the oscillations between the two partners takes place. Based on a mathematical model, the authors suggest a mechanism involving the oscillatory secretion/uptake of a signaling compound from a shared extracellular space.

---

## [Decision Letter]

**Decision letter after peer review:**

Thank you for submitting your article "Synchronization of oscillatory growth prepares fungal hyphae for fusion" for consideration by *eLife*. Your article has been reviewed by 3 peer reviewers, and the evaluation has been overseen by a Reviewing Editor and Naama Barkai as the Senior Editor. The following individual involved in review of your submission has agreed to reveal their identity: Antonio Di Pietro (Reviewer #1).

Essential revisions:

The reviewers' consensus is that the study's findings are valuable but that the strength of evidence provided is incomplete. It is essential that you:

1) Please increase the experiments' sampling sizes – the ones used are generally small, especially considering the observed variability.

2) The model was difficult to follow and added only incrementally to the model previously developed by Goryachev (Bioessays 2012). Please make its description clearer and clarify its novelty.

3) The calcium experiments were incomplete and do not fully demonstrate a role of extracellular calcium in hyphal communication. Please provide additional evidence that clearly demonstrates calcium's role.

4) Please improve the clarity of your figures and presented data – all reviewers had numerous constructive suggestions.

*Reviewer #1 (Recommendations for the authors):*

lines 111-139: I found the whole model section very confusing (see details below):

Lines 111-112: The term "compartments" as used here appears confusing. I suggest using the following sentence: "The concentration of signaling components in the immediate proximity of the wall of cell 1 and cell 2 is represented by *Z*1 and *Z*2, respectively".

Lines 111 and following: the authors need to clearly distinguish between signalling components (e.g. a cell receptor, a G protein, a protein kinase, etc.) or signalling compounds (ca^2+^, H2O2, peptides, etc.). I think here they mix both concepts. I guess this paragraph should read: "As a signaling compound is taken up into a cell, activating components (or compounds?) (modeled as circles *A*1 and *A*2) become more concentrated in this cell. These activating components (compounds?), in turn, stimulate the docking of cytoplasmic signaling components (*X*1 and *X* 2) to the compartment inside of the cell membrane (*Y*1 and *Y*2). The release of membrane-docked vesicles to the extracellular space is initially blocked by high levels of activator (*A*1 and *A*2). Secretion ultimately occurs when some of the activating components are converted into auto-inhibitory components (*I*1 and *I*2), reducing the levels of activating components, thereby allowing secretion of signaling compounds into the extracellular compartments (*Z*1 and *Z*2)"?

lines 49, 125, 127, 128, 134, 154, 177: should read "compound" instead of "component"

line 50. … depletion of ca^2+^ from the medium….

Line 136: use "implies" rather than "obviate"

Line 136 and following: Please clearly explain for the non-specialist why the model implies a slowdown of the oscillation dynamics. This is not obvious.

Figure 3b. Please explain in the legend what T1 and T2 mean (tip 1 and tip 2?).

Figure 3d. While the oscillations of GFP-ChsB are clearly visible in the images, those of R-GECO are not (at least to this reviewer's untrained eye). Although in the graph the R-GECO peaks are clear, to make the point the authors may want to try improving the image.

Line 165: I suggest that the authors use the word "possibly" rather than "probably". While the data demonstrate fluctuations of intracellular calcium, there is no evidence shown for the direct uptake of calcium during these rapid fluctuations, and thus they cannot exclude that the calcium could originate from intracellular stores.

Line 166 and subsequent: In the subsequent experiment, calcium was drastically depleted from the medium by addition of the calcium chelating agent EGTA for several hours. This led to reduced hyphal growth and abolished cell fusion. Moreover, fluorescence of R-GECO was not detectable in these ca^2+^-depleted conditions. The authors infer from this that "intracellular ca^2+^ oscillations indeed rely on extracellular ca^2+^". However, this conclusion is flawed because the hyphae have been depleted for calcium for several hours and therefore the intracellular stores are also likely depleted. Their data from EGTA treatment mainly confirm previous reports showing that calcium is required for hyphal fusion and also demonstrate that calcium is required for well-regulated pulsatile secretion and for the recruitment of SofT to the plasma membrane.

*Reviewer #2 (Recommendations for the authors):*

This paper uses live cell imaging to show the localization of a tagged protein, SofT, to hyphal tips during growth, independently of the oscillation of this protein during chemotropic interactions prior to cell fusion as shown in a published article by the authors in 2022 (Proc National Acad Sci). In addition, the role of calcium signaling is shown by using

Comments.

1. It is unclear if the constructs used in this study are the same as used in the PNAS 2022 paper. If so, it should be cited and not re-described. What is SofT? And how does the SofTnprotein function on a molecular level, if known.

2. SofT has been characterized in a number of fungi and oscillation of SofT during chemotropic interactions has been observed in a number of species. Is the oscillation of SoFT to hyphal tips unique in A. flagrans? and if so, why? It seems that the role of hyphae in exploring their environment is a universal feature. This aspect should be discussed.

3. There have been numerous published articles on the role of calcium in oscillatory growth of hyphae, most recent in 2022 (J. Fungi), including one by the author (PNAS 2017) and which are much more rigorous in approach.

4. As reported in the 2022 PNAS article, and elsewhere, cell dialog in a number of different fungi is mediated by a MAP kinase complex, which oscillates to the tip of interacting cells with opposing dynamics to SofT. This issue should be addressed, preferably by experimental data.

5. Are the oscillation dynamics of SofT similar or different to that of other fungi? If different, any ideas of the biochemical/mechanistic aspects of this? Are the oscillation dynamics temperature dependent (growth faster, faster dynamics)? Could it be that this oscillation is specific for nematode-trapping fungi? Does the presence of nematodes or nematode signaling molecules affect the dynamics of SofT?

6. The R-GECO oscillations are not really obvious from the panels or the movie.

7. Figure 4B is not clear and could be modified.

8. Is it possible that SofT is the only signaling component in this system? How would this change the model?

9. What is the evidence for the role of SofT playing a role in "exploration of its environment"? What is the phenotype of the SofT mutant and how would the oscillation and phenotype support this contention?

10. The role of calcium in the growth and chemotropic processes could be explored more thoroughly.

*Reviewer #3 (Recommendations for the authors):*

General points:

Dedicated introduction and Discussion sections would improve the flow of the manuscript. There is a very limited discussion to contextualize this work in a larger context of cell to cell communication, membrane fusion or even the realm of coupling of biochemical oscillation. Ideally, signaling molecule(s) would be identified through experiments, but at the very least there should be a Discussion section postulating on possibilities on the possible nature of signals. Perhaps the mathematical model could be used to put limits on diffusion/secretion rates that would support the observed oscillatory behavior and could point to an ion vs macromolecule based signal?

The N's are small in general (ranging from 3-12 hyphae) and it is difficult to grasp the cell to cell variability in the oscillations. With SD of 30%, it seems there is substantial variability. Experiments that show the degree to which a given cell monologue persists at a given frequency vs fluctuating would be interesting and maybe useful to compare as a function of distance to a partner to understand the transition from monologue to out of synchrony and dialogue.

Figure 1:

– This sentence could be more clear: "Numbers indicate the count of oscillating accumulation of each fusion protein at the hyphal times during the time course."

– The peak at 60s in the ChsB histogram is perplexing because there are not clear peak-to-peak intervals of 60s. Is the plot in C not representative? Could you be more clear with how the signals are being processed and analyzed?

– Is the ChsB signal plotted with fewer data points than the SofT signal? Also, the kymograph in 1B appears to be blurry

Figure 2:

– Could you make it more clear where (if anywhere) ChsB and SofT fit in the mathematical model?

Figure 3: Could synchronization of the signals in B be quantified? It would be great to have a plot that includes data from more than just a single pair of hyphae to show that this pattern of asynchrony to synchrony is a frequent occurrence. Additionally, as noted above, it would be interesting to see how the oscillations change as a frequency of distance and starting point frequencies of the hyphae in the dialogue.

Figure 4: do you have any experimental evidence for the so-called refractory period that is preventing self stimulation?

If there was any way to either manipulate the oscillations to force to be in phase, potentially through local light induced recruitment of soft or in some way demonstrate that out of phase really matters or linking calcium as the direct signal, this would make all the difference in making this paper more of an advancement for the field.

Could an experiment be conceptualized that somehow assesses the extracellular role of calcium as a possible signal that is conveying the antiphase information? I wonder if there are any extracellular dyes or even recombinantly supplied proteins that could be taken advantage of to image calcium, potentially in the context of small growth volumes or microfluidics? Are there ways to more transiently, locally chelate calcium between two homing tips, maybe with a photocaged chelation?

---

## [Author Response]

Essential revisions:The reviewers' consensus is that the study's findings are valuable but that the strength of evidence provided is incomplete. It is essential that you:1) Please increase the experiments' sampling sizes – the ones used are generally small, especially considering the observed variability.

We increased sample sizes and updated the respective graphs. To further validate our findings, we included measurements of the MAP kinase MakB (MAK-2) and the actin marker Lifeact (see new Figure 3 and Figure S5)

2) The model was difficult to follow and added only incrementally to the model previously developed by Goryachev (Bioessays 2012). Please make its description clearer and clarify its novelty.

To clarify the novelty of our model extension, and relate our model better to the biological system, we added the following in the model section:

“The main novelty of our model is the addition of the variables Z_1_ and Z_2_ to explicitly represent the information exchange between hyphae via the shared extra-cellular medium. Among the abstract model variables, SofT as a signaling component located inside of a hyphal cell connects most closely to variables X1,2 and Y1,2. ChsB is also located inside the hyphal cells, but with localizes differentlyfrom SofT and likely does not act as a signalling component, so that ChsB can be related to the variables A1,2, and I1,2.”

We further modified Figure 2A (model schematic) and revised the text on the model construction and model analysis based on the input from all three referees.

3) The calcium experiments were incomplete and do not fully demonstrate a role of extracellular calcium in hyphal communication. Please provide additional evidence that clearly demonstrates calcium's role.

To provide additional evidence, we concentrated on proteins which potentially play a role in coupling the ca^2+^ signal into hyphal growth and the cell dialogue.

First, we focused on orthologues of the protein FIG1 (mating factor induced gene 1), which is part of the low-affinity calcium uptake system (LACS) in *Saccharomyces cerevisiae* and is essential for mating. Interestingly, filamentous fungi like *A. flagrans* contain two orthologues in their genome and we identified *A. flagrans* FigA and FigB. Single gene deletion of *figA* resulted in a colony phenotype, but both gene deletion mutants still performed cell fusions and trapped nematodes (Figure S9). We were unable to generate a double deletion, however, this might have been since at least one protein is essential for general growth. Indeed, in a recent publication (Zhao et al., 2023) the orthologue DhFIG_2 of *Dactylellina haptotyla* could only be knocked down by RNA interference. Considering that the MAP kinase MAK-2 (MakB) plays an essential role in vegetative hyphal fusion in filamentous fungi, while the MakB-orthologue Fus3 is essential for mating, we believe that a direct comparison between mating and hyphal fusion might be helpful identifying novel components of the system. We added parts of our results to the discussion and created a new figure (Figure S9).

Interestingly, the Soft protein sequence exhibits weak homology to mammalian proteins of the aczonin/piccolo family (Goryachev et al., 2012). These proteins act as scaffold proteins in the active zone of synapses to regulate the coordinated fusion and release of vesicles and their content into the synaptic cleft. Since the neurotransmitter release is highly dependent on calcium, we screened for Soft (PRO40)-interacting proteins with calcium-dependent functions in the filamentous fungus *Sordaria macrospora*, as published by Teichert et al., 2014. We identified HAM-10, an orthologue of Unc13/Munc13 in higher eukaryotes. Unc13/Munc13 is known to facilitate proper SNARE assembly at the vesicle release site in the synaptic cleft (Dittman, 2013). The *A. flagrans* orthologue HamA contains a calcium dependent C2 domain, similar to orthologues in other fungi. In higher eukaryotes, the protein also contains a calmodulin-binding domain. To further investigate the role of HamA, we generated a *∆hamA* deletion strain, which showed a cell fusion mutant phenotype as observed in the ∆*ham-10* mutant in *N. crassa* (Fu et al., 2011) (Figure S10). Interestingly, in the *∆ham-10* mutant, neither Soft nor MAK-2 localized at the tip of germ tubes (Fu et al., 2014). We were only able to localize mCherry-ChsB in a *∆hamA* mutant background and observed oscillating recruitment of the protein to the hyphal tip, indicating that HamA could play a role at the interface of combining cell dialogue signaling and growth dynamics.

One interacting protein of Munc13 is RIM1, which plays a role in linking vesicle fusion and calcium influx. We identified the orthologue in the Soft/PRO40 interacting protein dataset of Teichert et al., 2014. Similarly, the orthologous Syt1 in *N. crassa* is involved in cell fusion, but not essential (Palma-Guerrero et al., 2014). We localized the orthologues RimA in *A. flagrans* which shows localization at the hyphal tip without any noticeable oscillations during our time series (Figure S10).

Additionally, calmodulin and a calcium/calmodulin-dependent protein kinase were identified as interaction partner of PRO40, indicating strong similarities between the fast and highly regulated secretion of the cell dialog in fungi and release of synaptic vesicles in neurons. Early neural cell-type can be observed in ctenophores, which are one of the earliest-branching lineages of bilaterians who share some neuronal genes (Burkhardt and Jékeley, 2021). In a so-called Unc13-RIM line, the combination of voltage-gated calcium channels to the vesicle release machinery results in coupling of action potentials and calcium influx to swift vesicle release. These results further show the occurrence of pulsatile secretion across eukaryotes. In the future, it will be interesting to study how other fusion-related proteins might be involved in the interpretation of the increase in intracellular ca^2+^-concentrations and how this change relates to the secretion of a putative chemo-attractive signal molecule. Our data add evidence to the hypothesis, that extracellular calcium plays a role in hyphal communication, however, we feel that deciphering the exact role during this process might be out of the scope of this short report.

4) Please improve the clarity of your figures and presented data – all reviewers had numerous constructive suggestions.

We improved the clarity of our figures and included additional experiments suggested by the reviewers.

The following experiments were added to help strengthening our findings:

We quantified the occurrence of SofT-GFP monolog on LNA and observed the same behavior on PDA. We included a new figure (Figure S1).We performed rescue experiments of the ∆*sofT* mutant (Youssar et al., 2019) with truncated fragments of the protein and found that the N-terminal disordered region and the WW domain of the protein are sufficient to rescue the cell fusion deficiency of the mutant. Our results are summarized in (Figure S2).We included additional microscopy data showing that hyphae without SofT-GFP recruitment at their tips were unable to fuse, suggesting that they could not recognize each other. (new Movie S7) and new figure Figure S1.We localized the SofT orthologue in *A. nidulans* and observed no monolog in this species under our tested conditions (on *Aspergillus* minimal media (solid/liquid), LNA,). We hypothesize that depending on the species there might be fundamental difference if a hyphal cell undergoes a signaling monologue and discuss this result in the manuscript.We localized the MAP kinase MakB and co-localized it with SofT. We discovered that both proteins show oscillations in the same phase without other hyphae in vicinity, which is the opposite of the so far observed anti-phasic oscillations observed during the cell dialogue (Fleissner et al., 2009; Hammadeh et al., 2022). Additionally, we observed that decoupling of both oscillations into the anti-phasic cell dialogue occurred during the transitory phase where the growth slowed down. We feel that this observation is a strong contribution to our findings and therefore split our previous figure 3 into new figures 3 and 4, separating our transition findings (new Figure 3) and Calcium-related (new Figure 4) findings.We improved the visualization of R-GECO fluorescence and included previous Figure S2b into new Figure 4a for further clarification. We additionally measured actin dynamics visualized by Lifeact-GFP during the cell dialogue and included our findings in new Figure S5b.We quantified SofT-GFP oscillations in the presence of the nematode *C. elegans* and couldn’t observe changes in the dynamic.

Reviewer #1 (Recommendations for the authors):lines 111-139: I found the whole model section very confusing (see details below):

This sentence has been changed as suggested.

Lines 111-112: The term "compartments" as used here appears confusing. I suggest using the following sentence: "The concentration of signaling components in the immediate proximity of the wall of cell 1 and cell 2 is represented by Z1 and Z2, respectively".Lines 111 and following: the authors need to clearly distinguish between signalling components (e.g. a cell receptor, a G protein, a protein kinase, etc.) or signalling compounds (ca^2+^, H2O2, peptides, etc.). I think here they mix both concepts. I guess this paragraph should read: "As a signaling compound is taken up into a cell, activating components (or compounds?) (modeled as circles A1 and A2) become more concentrated in this cell. These activating components (compounds?), in turn, stimulate the docking of cytoplasmic signaling components (X1 and X2) to the compartment inside of the cell membrane (Y1 and Y2). The release of membrane-docked vesicles to the extracellular space is initially blocked by high levels of activator (A1 and A2). Secretion ultimately occurs when some of the activating components are converted into auto-inhibitory components (I1 and I2), reducing the levels of activating components, thereby allowing secretion of signaling compounds into the extracellular compartments (Z1 and Z2)"?

See explanation below after next comment…

lines 49, 125, 127, 128, 134, 154, 177: should read "compound" instead of "component"

lines 49, 125, 127, 128, 134, 154, 177: should read "compound" instead of "component" We agree with the referee’s wish to assign the more concrete terms of compound / component, or even specific molecular species for the model variables. The reason we did not do this in the original submission is simply that it is not fully known what these components are. It is, in our perception, a strength of our model that this molecular specificity is in fact not required to still make very specific predictions that are also reflected in the experiments. Instead, the model suggests what “rules” or “behaviors” the unknown components must represent. This is a common approach in these types of models, and many in our community see it as one of the main advantages of model building that these theories can guide the further design of experiments – as indeed, we do in the rest of the study, following our model analysis. It is then in the nature of this approach that we cannot clearly say whether we are dealing with a compound (monomeric macromolecular species or even a simple chemical) or with a cellular component (proteins). We therefore assign the agnostic term of “component” throughout, to not give more concrete naming than would be warranted by our model.

line 50. … depletion of ca^2+^ from the medium….

Corrected.

Line 136: use "implies" rather than "obviate"

Corrected.

Line 136 and following: Please clearly explain for the non-specialist why the model implies a slowdown of the oscillation dynamics. This is not obvious.

The referenced “critical slowing down” is not a property of this particular model, but rather a phenomenon that is observed for most dynamical systems that go from one dynamical behavior into another type of behavior, as one main model parameter is modified while the system keeps running. We tried to reformulate the according sentence as below. Any longer discussion would require the study of textbooks or additional articles, as the underlying mechanisms of this slowing down are well-known in the dynamical systems, but are not trivial to explain in terms of their dynamic original. We refrained from such explanations, and instead hope that the following explanation gives an intuition, and supply the according references at the end of the sentence:

This “critical-slowing down” is typical in phases where a system transitions from one type of dynamic behavior to another type of dynamic behavior, due to being “caught in between” two types of behavior during the transition (1, 23).

Figure 3b. Please explain in the legend what T1 and T2 mean (tip 1 and tip 2?).

Yes, we included the information in the figure legend.

Figure 3d. While the oscillations of GFP-ChsB are clearly visible in the images, those of R-GECO are not (at least to this reviewer's untrained eye). Although in the graph the R-GECO peaks are clear, to make the point the authors may want to try improving the image.

We displayed the images of 3d (now new Figure 4a) in greyscale (inverted contrast) and used different frames of the time series to help with visibility. We additionally included previous Figure S2b into the new Figure 4a to further help with clarification.

Line 165: I suggest that the authors use the word "possibly" rather than "probably". While the data demonstrate fluctuations of intracellular calcium, there is no evidence shown for the direct uptake of calcium during these rapid fluctuations, and thus they cannot exclude that the calcium could originate from intracellular stores.

Corrected.

Line 166 and subsequent: In the subsequent experiment, calcium was drastically depleted from the medium by addition of the calcium chelating agent EGTA for several hours. This led to reduced hyphal growth and abolished cell fusion. Moreover, fluorescence of R-GECO was not detectable in these ca^2+^-depleted conditions. The authors infer from this that "intracellular ca^2+^ oscillations indeed rely on extracellular ca^2+^". However, this conclusion is flawed because the hyphae have been depleted for calcium for several hours and therefore the intracellular stores are also likely depleted. Their data from EGTA treatment mainly confirm previous reports showing that calcium is required for hyphal fusion and also demonstrate that calcium is required for well-regulated pulsatile secretion and for the recruitment of SofT to the plasma membrane.

We agree with this comment and changed our conclusion regarding the EGTA treatment.

Reviewer #2 (Recommendations for the authors):This paper uses live cell imaging to show the localization of a tagged protein, SofT, to hyphal tips during growth, independently of the oscillation of this protein during chemotropic interactions prior to cell fusion as shown in a published article by the authors in 2022 (Proc National Acad Sci). In addition, the role of calcium signaling is shown by usingComments.1. It is unclear if the constructs used in this study are the same as used in the PNAS 2022 paper. If so, it should be cited and not re-described. What is SofT? And how does the SofTnprotein function on a molecular level, if known.

The *sofT*-GFP constructs created in this study were designed to tag the gene in locus and transformed into *A. flagrans* WT. In the PNAS 2022 paper, a ∆*sofT* mutant strain was complemented with a GFP-*sofT* construct which integrated ectopically in the genome. The reasoning behind it was because of the limited number of selection markers in *A. flagrans*.

The Soft protein is only conserved in filamentous ascomycete fungi. The *Neurospora crassa* and *Sordaria macrospora* Soft-orthologues proteins SO and PRO40 are required for sexual development. The orthologue protein AoSO of *Aspergillus oryzae* localizes at septal pores upon damage to hyphal compartments and various stresses like temperature, pH or nutrient depletion.

Soft-orthologues contain 1200-1300 amino acids with a conserved N-terminal disordered region and a central WW domain, which facilitates protein-protein interactions in signaling networks. It was shown that the *S. macrospora* PRO40 acts as a scaffold protein of the cell wall integrity pathway, however its molecular function remains insufficiently understood (Teichert et al., 2014).

In order to further dissect its molecular role, we generated three truncated versions of the *A. flagrans* SofT protein and performed rescue experiments of the ∆*sofT* mutant and included our results in Figure S2. *A. flagrans* SofT contains 1213 amino acids. Analysis with the NCBI domain prediction tool coupled with the iupred2a prediction tool (https://iupred2a.elte.hu) predicted an N-terminal disordered region, a WW domain (AA505-533) and a C-terminal putative phosphatase domain (AA676-1213) in the sequence (Figure S2a). The first fragment in our rescue experiment encoded for the N-terminal region of the protein (1-504 AA) without the conserved WW domain. The second fragment encoded for the N-terminal region with the conserved WW domain (1- 540 AA). The third fragment encoded the C-terminal region of the protein containing the putative phosphatase domain (534 – 1213 AA). We individually transformed the three constructs into the ∆*sofT* strain and observed the growth of the mutants after genotyping. As the ∆*sofT* deletion shows a strong growth phenotype on solid media (Youssar et al., 2019) with the absence of aerial mycelium, we checked the growth of the three mutants on potato dextrose agar. The ability to form aerial mycelia and perform vegetative hyphal fusion was only restored by the second fragment containing the N-terminal region with the WW domain (Figure S2b, c) suggesting the importance of SofT-interacting proteins during the cell dialog.

2. SofT has been characterized in a number of fungi and oscillation of SofT during chemotropic interactions has been observed in a number of species. Is the oscillation of SoFT to hyphal tips unique in A. flagrans? and if so, why? It seems that the role of hyphae in exploring their environment is a universal feature. This aspect should be discussed.

Parts of this response were included in the Results section, resulted in a new figure (Figure S1) and were discussed in.

We quantified the presence of SofT at 3x50 hyphal tips in *A. flagrans* hyphae on LNA in order to characterize monolog oscillations. We found that 85% +- 4 (average +-SD) showed localization of SofT at the hyphal tip in the absence of a fusion partner. We also observed that hyphae without SofT-GFP recruitment at their tips were unable to fuse, suggesting that they could not recognize each other (Figure S1b). However, these cells were still able to undergo cell fusion if other hyphae induced hyphal fusion. We also tested the SofT-GFP dynamics on PDA (rich nutrient media compared to low nutrient LNA) and observed recruitment to the hyphal tips and cell fusion, suggesting that nutrient availability would not be the sole cause of inducing the dynamics (Figure S1c)

We additionally identified and localized the SofT orthologue in *Aspergillus nidulans* to check if oscillation of SofT at hyphal tips can be found in different fungi. We found punctate localization throughout the hyphae which resembles the punctuate localization that can be also observed in *A. flagrans* (see Figure S8 and Figure S1a, c for a comparison) but couldn’t observe any tip oscillation of *A. nidulans* SO during cultivation in liquid and on solid *Aspergillus* minimal medium as well as low nutrient agar as it is used in *A. flagrans* experiments. However, we also observed no hyphal fusion during these culture conditions suggesting that oscillations of Soft at hyphal tips might be related to the preparedness of hyphal fusion. Interestingly, *Aspergillus* species are known to rarely undergo vegetative hyphal fusion under laboratory conditions (Macdonald et al., 2018), while it can be observed more frequently in other fungi. We speculate that at least under experimental conditions there might be ways to regulate/activate the signaling monologue and it might be different in fast growing fungi (in molds like *Aspergillus* or *Neurospora*) compared to slower growing fungi which will be exciting studying in the future.

3. There have been numerous published articles on the role of calcium in oscillatory growth of hyphae, most recent in 2022 (J. Fungi), including one by the author (PNAS 2017) and which are much more rigorous in approach.

See our general response 3 to the editor for further clarification. In the study performed by Kurian et al., (2022), evidence of ca^2+^ signaling during cell fusion was presented by using various ca^2+^ pharmacological modulators. We believe that our contribution of showing that the stepwise extension of hyphae is coordinated during the cell-to-cell dialogue and which we visualized by pulses of intracellular ca^2+^ accumulations include a new angle of showing the involvement of ca^2+^ during the process.

4. As reported in the 2022 PNAS article, and elsewhere, cell dialog in a number of different fungi is mediated by a MAP kinase complex, which oscillates to the tip of interacting cells with opposing dynamics to SofT. This issue should be addressed, preferably by experimental data.

Thank you for the great suggestion. We localized the MAP kinase MakB fused to mCherry during hyphal growth of *A. flagrans*. MakB-mCherry showed oscillatory recruitment to nuclei of growing hyphae with phases similar to the observed SofT-GFP oscillations. Interestingly, co-localization of SofT-GFP and MakB-mCherry in the same hyphae revealed that both proteins were oscillating in the same phase without other hyphae in vicinity, which is the opposite of the so far observed anti-phasic oscillations observed during the cell dialogue. Additionally, we observed that decoupling of the oscillations into the anti-phasic cell dialogue occurred during the transitory phase. We included our results in (L167) and updated the figures to create a new figure 3 and supplementary figure Figure SS.

5. Are the oscillation dynamics of SofT similar or different to that of other fungi? If different, any ideas of the biochemical/mechanistic aspects of this? Are the oscillation dynamics temperature dependent (growth faster, faster dynamics)? Could it be that this oscillation is specific for nematode-trapping fungi? Does the presence of nematodes or nematode signaling molecules affect the dynamics of SofT?

So far, the oscillation dynamics of SofT/MakB have been reported in *Neurospora crassa*, *Botrytis cinerea*, *and A. flagrans*, exhibiting minute time scales that vary across these species. Specifically, in *N. crassa*, the oscillation occurs approximately every 10 minutes, while *in B. cinerea*, the range extends from 11 to 15 minutes (Fleissner et al., 2009; Hammadeh et al., 2022; Fleissner et al., 2023). It is important to note that these observations were conducted on different media, which may have an impact on the dynamics. Additionally, the growth speed and size of all tested fungi are different as well. Considering the coupling between SofT oscillation and hyphal growth, we speculate that the overall dynamics of the oscillation will be influenced by the speed of growth. We tested if SofT-GFP oscillations are modified by nematodes and could not observe any differences. We haven’t checked for changes at different temperatures but speculate that during colder incubations (when growth slows down) oscillations of SofT might slow down, too. As seen in our comment 2 we didn’t observe oscillation dynamics of Soft *in A. nidulans* but agree, that this observation opens new questions about whether and why some fungi show this behavior while others don’t. Finding media conditions that either repress or induce hyphal fusion and comparing different fungal species might bring new insights into the observed dynamics.

6. The R-GECO oscillations are not really obvious from the panels or the movie.

We displayed the images of 3d in greyscale (inverted contrast) and used different frames of the time series to help with visibility. We additionally included previous Figure S2b into the new Figure 4a to further clarify.

7. Figure 4B is not clear and could be modified.

We modified the scheme. It is now new Figure 5.

8. Is it possible that SofT is the only signaling component in this system? How would this change the model?

This is a justified question, which has been addressed by previous work on this type of oscillatory system (Groyachev et al. 2012, see below). We expanded on this point in our model section, as follows:

“This model requires that not one, but two species in the cell undergo a conversion, with the second species’ conversion requiring the first species’ transition to have completed to a large extent. The reason behind this is that, without this sequential conversion, there is no delay between receiving and secreting signal, and that the secreting cell would not become nonreceptive to its own signal (8).”

(8) A. B. Goryachev, A. Lichius, G. D. Wright, N. D. Read, Excitable behavior can explain the "ping-pong" mode of communication between cells using the same chemoattractant. *Bioessays* 34, 259-266 (2012).

9. What is the evidence for the role of SofT playing a role in "exploration of its environment"? What is the phenotype of the SofT mutant and how would the oscillation and phenotype support this contention?

Deletion of *sofT* results in absence of hyphal cell fusion in the mycelia (see Figure S2c and Youssar et al., 2019). As seen in comment 2 (Figure S1b) we observed that hyphae without SofT-GFP recruitment at their tips were unable to fuse, suggesting that they could not recognize each other.

10. The role of calcium in the growth and chemotropic processes could be explored more thoroughly.

See our general response 3 to the editor for further clarification. We believe that our contribution of showing that the stepwise extension of hyphae is coordinated during the cell-tocell dialogue and which we visualized by pulses of intracellular ca^2+^ accumulations include a new angle of showing the involvement of ca^2+^ during the process.

Reviewer #3 (Recommendations for the authors):General points:Dedicated introduction and Discussion sections would improve the flow of the manuscript. There is a very limited discussion to contextualize this work in a larger context of cell to cell communication, membrane fusion or even the realm of coupling of biochemical oscillation. Ideally, signaling molecule(s) would be identified through experiments, but at the very least there should be a Discussion section postulating on possibilities on the possible nature of signals. Perhaps the mathematical model could be used to put limits on diffusion/secretion rates that would support the observed oscillatory behavior and could point to an ion vs macromolecule based signal?

We combined results and discussion as our findings were thought to be published as short report and feel that this format gives more strength to our observations.

Since the new concept of *eLife* was launched meanwhile we are unsure about the format. Therefore, we updated the format.

As seen in our general response 3 to the editor we see strong similarities between the hyphal cell dialogue and neurotransmitter release in the synaptic cleft including membrane fusion and added this as additional discussion in the manuscript. Coupling of biochemical oscillations are certainly possible and indeed 6-phosphofructo-2-kinase was identified as phosphorylation target protein of MAK-2 in *N. crassa* (Jonkers et al., 2014) and direct interacting protein of Soft in *S. macrospora* (Teichert et al., 2014), suggesting a link between cell dialogue and the glycolytic oscillation which needs further investigation. In addition,

We added a new section "Particle size estimation for the signaling component" to the Methods. Based on this estimation, we now added the following the main manuscript text:

“An estimate calculation indicates that particles of diameter of approximately 3 nm or less can traverse the distance at which two hyphae synchronize at a sufficiently short time to support synchronization (for details, see Methods). This diameter would include ca^2+^ ions as well as ca^2+^binding proteins, but not secretory vesicles as the component that is exchanged between the hyphae.”

The N's are small in general (ranging from 3-12 hyphae) and it is difficult to grasp the cell to cell variability in the oscillations. With SD of 30%, it seems there is substantial variability. Experiments that show the degree to which a given cell monologue persists at a given frequency vs fluctuating would be interesting and maybe useful to compare as a function of distance to a partner to understand the transition from monologue to out of synchrony and dialogue.

We increased sample sizes and updated the respective graphs. To further validate our findings, we included measurements of the MAP kinase MakB (MAK-2) and the actin marker Lifeact. (see Figure 3, Figure S4 and Figure S5).

Figure 1:– This sentence could be more clear: "Numbers indicate the count of oscillating accumulation of each fusion protein at the hyphal times during the time course."

Corrected.

– The peak at 60s in the ChsB histogram is perplexing because there are not clear peak-to-peak intervals of 60s. Is the plot in C not representative? Could you be more clear with how the signals are being processed and analyzed?

Each histogram is the result of a measurement of the fluorescent intensity with a circular selection at the hyphal tip for every given frame. The relative fluorescent intensity values are then plotted against the time as depicted in Figure 1c. The plot in C is representative for the time series shown in Figure 1b. We are not clearly sure about the first part of this comment since there is a fluorescent accumulation (peak) at 60s.

– Is the ChsB signal plotted with fewer data points than the SofT signal? Also, the kymograph in 1B appears to be blurry

Yes, the mCherry-ChsB images were captured at reduced intervals (typically at 30seconds) due to quicker photobleaching.

Figure 2:– Could you make it more clear where (if anywhere) ChsB and SofT fit in the mathematical model?

We added an according extension to Figure 2B, indicatint how these protein species might figure in the model. Further, we added the following to the text of our manuscript:

“Among the abstract model variables, SofT as a signaling component located inside of a hyphal cell connects most closely to variables X_1,2_ and Y_1,2_ (Figure 2a). ChsB is also located inside the hyphal cells, but localizes differently from SofT and likely does not act as a signaling component, so that ChsB can be related to the variables A_1,2_, and I_1,2_.”

Figure 3: Could synchronization of the signals in B be quantified? It would be great to have a plot that includes data from more than just a single pair of hyphae to show that this pattern of asynchrony to synchrony is a frequent occurrence. Additionally, as noted above, it would be interesting to see how the oscillations change as a frequency of distance and starting point frequencies of the hyphae in the dialogue.

We additionally localized the MAP kinase MakB and co-localized it with SofT. We discovered that both proteins show oscillations in the same phase without other hyphae in vicinity, which is the opposite of the so far observed anti-phasic oscillations observed during the cell dialogue (Fleissner et al., 2009; Hammadeh et al., 2022). Additionally, we observed that decoupling of both oscillations into the anti-phasic cell dialogue occurred during the transitory phase and included these results in Figure 3.

Figure 4: do you have any experimental evidence for the so-called refractory period that is preventing self stimulation?

So far the refractory period is based on models of excitable behavior. Of note, the fusing cells are genetically identical cells and hence would produce the same putative chemoattractant molecule to recognize the partner.

If there was any way to either manipulate the oscillations to force to be in phase, potentially through local light induced recruitment of soft or in some way demonstrate that out of phase really matters or linking calcium as the direct signal, this would make all the difference in making this paper more of an advancement for the field.

This is an interesting thought, and we agree that decoupling components of the system will be one of the next steps forward in the field. However, we believe that our discoveries of a monologue of individual hyphae before fusion, the transition into a dialogue and the coordination of growth during the cell-to-cell dialogue provide new valuable information for the field.

Could an experiment be conceptualized that somehow assesses the extracellular role of calcium as a possible signal that is conveying the antiphase information? I wonder if there are any extracellular dyes or even recombinantly supplied proteins that could be taken advantage of to image calcium, potentially in the context of small growth volumes or microfluidics? Are there ways to more transiently, locally chelate calcium between two homing tips, maybe with a photocaged chelation?

Great idea! Observing cell dynamics in microfluidic chambers and manipulating will be something to consider for the future.